# The effects of a deleterious mutation load on patterns of influenza A/H3N2's antigenic evolution in humans

Katia Koelle[1,2]*, David A Rasmussen[1,3]

[1]Department of Biology, Duke University, Durham, United States; [2]Fogarty International Center, National Institutes of Health, Bethesda, United States; [3]Department of Biosystems Science and Engineering, Eidgenössische Technische Hochschule Zürich, Basel, Switzerland

**Abstract** Recent phylogenetic analyses indicate that RNA virus populations carry a significant deleterious mutation load. This mutation load has the potential to shape patterns of adaptive evolution via genetic linkage to beneficial mutations. Here, we examine the effect of deleterious mutations on patterns of influenza A subtype H3N2's antigenic evolution in humans. By first analyzing simple models of influenza that incorporate a mutation load, we show that deleterious mutations, as expected, act to slow the virus's rate of antigenic evolution, while making it more punctuated in nature. These models further predict three distinct molecular pathways by which antigenic cluster transitions occur, and we find phylogenetic patterns consistent with each of these pathways in influenza virus sequences. Simulations of a more complex phylodynamic model further indicate that antigenic mutations act in concert with deleterious mutations to reproduce influenza's spindly hemagglutinin phylogeny, co-circulation of antigenic variants, and high annual attack rates.

*For correspondence: katia.koelle@duke.edu

Competing interests: The authors declare that no competing interests exist.

## Introduction

Seasonal influenza viruses infect up to 15% of the world's human population annually, with the majority of flu cases attributable to influenza type A subtype H3N2 (A/H3N2) (*World Health Organization, 2014*). This substantial disease burden stems from the virus's rapid antigenic evolution, which enables it to infect hosts within several years of a previous infection. A large body of research has therefore focused on understanding the process by which influenza evolves antigenically, particularly how point mutations in the virus's hemagglutinin (HA) protein allow for immune escape (*Wiley et al., 1981*; *Wilson and Cox, 1990*; *Koel et al., 2013*) and how virus strains interact immunologically to shape this subtype's evolutionary patterns in the long term (*Ferguson et al., 2003*; *Tria et al., 2005*; *Koelle et al., 2006*; *Recker et al., 2007*; *Bedford et al., 2012*; *Zinder et al., 2013*).

Distinct from these efforts, several phylogenetic analyses have indicated that influenza A/H3N2 in humans carries a deleterious mutation load (*Fitch et al., 1997*; *Pybus et al., 2007*; *Strelkowa and Lässig, 2012*). Specifically, early work by *Fitch et al. (1997)* found the number of nonsynonymous changes on tip branches of the HA phylogeny to be higher than expected, indicative of either strain selection bias or the presence of transiently circulating deleterious mutations in the influenza viral population. In more recent work, *Pybus et al. (2007)* performed a comprehensive phylogenetic analysis of over 140 viruses, including influenza A/H3N2. For H3N2's M1 protein, as well as for the majority of the other viral proteins examined in the study, they found heightened ratios of non-synonymous-to-synonymous substitutions on external tree branches relative to those found internally. This finding again points towards transiently circulating deleterious mutations in influenza and, more

**eLife digest** Each year, up to 15% of the world's population experience symptoms of an influenza infection, also commonly known as flu. The most common culprit is a strain of the virus called influenza type A subtype H3N2. One reason that so many people become infected each year is that this virus evolves rapidly. Within a few years, proteins on the surface of the virus known as antigens become less recognizable to the immune system of a person who has been previously infected. This means that the person can become ill with the virus again because their immune system cannot mount an effective response to the evolved virus strain.

Influenza virus strains evolve rapidly because their genetic material accumulates mutations quickly. Although some of these mutations are beneficial to the virus, other mutations are harmful and reduce the ability of the virus to spread. Sometimes beneficial mutations may occur alongside harmful ones, but it is not known how the harmful mutations affect the evolution of the virus.

Here, Koelle and Rasmussen used computer models of H3N2 influenza to examine the effect of harmful mutations on the evolution of this virus population. The models show that harmful mutations limit how quickly the antigens can evolve. Also, the presence of these harmful mutations effectively acts as a sieve: they allow only large changes in the antigens to establish in the virus population.

The models suggest that there are three routes by which large changes in the antigens on H3N2 viruses may occur. The first is by a single mutation that has a big effect on the antigens in viruses that only carry a few harmful mutations, but these large mutations would not happen very often. Another route may be through more common mutations that have only a small or moderate benefit, which would allow the virus to become more common in the population before it acquires a beneficial mutation with a much greater effect. The third possibility is that a large beneficial mutation may arise in viruses that have many harmful mutations. These harmful mutations may initially limit the ability of the virus to spread, but over time, some of these harmful mutations may then be lost.

Koelle and Rasmussen found that the computer models could recreate the patterns of virus evolution that have been observed in real strains of H3N2. Researchers use predictions of influenza evolution to help them decide which virus strains should be included in flu vaccines each year. Koelle and Rasmussen findings indicate that harmful mutations should be considered when making these predictions.

generally, across RNA virus populations. Other recent work on predicting the short-term evolution of influenza has highlighted the necessity of accounting for fitness costs associated with sublethal deleterious mutations when projecting the frequencies of influenza clades into the next season (*Łuksza and Lässig, 2014*). Together, these results indicate that purifying selection is not sufficiently strong to immediately eliminate deleterious mutations from the influenza A/H3N2 virus population that circulates among humans. As a result of genetic linkage within genes and, to a lesser extent, across genes, these deleterious mutations have the potential to interact with beneficial mutations in determining which viral lineages will persist and which ones will ultimately be lost. Indeed, a recent statistical analysis of HA sequences from influenza A/H3N2 has suggested that interference effects largely determine the fates of viral mutants, rather than their inherent selective effects (*Illingworth and Mustonen, 2012*). These interference effects are possible because of an extensive genetic linkage across influenza's HA (*Strelkowa and Lässig, 2012*) and arise from variation in the background fitness of viral strains and from variation in the fitness effects of subsequent mutations.

Taken together, this body of work indicates that sublethal deleterious mutations commonly arise and circulate for sufficiently long periods of time to be able to impact the trajectories of influenza A/H3N2 strains. However, the impact that these deleterious mutations have on the population dynamics and long-term evolutionary patterns of this subtype has to date not been explored. Here, we address this question with a set of increasingly complex population genetic and population dynamic models, under the common assumption that influenza's adaptive evolution is driven by antigenic changes that allow for escape from herd immunity. We start by extending classic population genetic models into an epidemiological context. As expected from previous analyses of these types of models (*Fisher, 1930*; *Birky and Walsh, 1988*; *Peck, 1994*; *Barton, 1995*; *Orr, 2000*), we find that circulating sublethal deleterious mutations in influenza A/H3N2's viral population reduce the rate of

adaptive evolution and increase the average size of the beneficial mutants that fix. Extending this analysis to models explicitly incorporating epidemiological dynamics, we further show that the accumulation of deleterious mutations can contribute to explaining the invasion dynamics of new antigenic clusters, defined as sets of viral strains that are antigenically similar to one another (*Smith et al., 2004*). This model further predicts three distinct molecular pathways by which antigenic cluster transitions can occur, and we find empirical patterns consistent with each of these pathways in sequence data from 1992 to 2004.

Gaining intuition from these simple models, we then present more extensive phylodynamic simulations that incorporate the occurrence of both antigenic and non-antigenic (largely deleterious) mutations. This extension is critical given that antigenic mutations acquire their selective advantage through immune escape, which reduces competition for susceptible hosts and therefore in principle could allow for long-term coexistence of virus strains through niche partitioning of the host population (*Cobey, 2014*). Indeed, in the absence of other contributing processes, it has been shown that reduced between-strain competition resulting from antigenic evolution leads to explosive genetic and antigenic diversity (*Ferguson et al., 2003*), a pattern that is inconsistent with the long-term evolutionary dynamics of influenza A/H3N2 in humans. Intriguingly, the phylodynamic model we present robustly reproduces the spindly phylogeny of influenza A/H3N2's HA protein (*Fitch et al., 1997*) under parameterizations relevant to A/H3N2 in humans. It further reproduces the recently described patterns of co-circulation of minor antigenic variants (*Strelkowa and Lässig, 2012*), as well as high annual attack rates (*World Health Organization, 2014*). In the discussion, we situate these findings in the context of previously published models used to explain the characteristic evolutionary dynamics of influenza A/H3N2 in humans, speculate on the applicability of these findings to other influenza-host systems, and comment on the consequences of these findings for the predictability of influenza evolution.

## Results

### A deleterious mutation load modifies the tempo and nature of influenza's antigenic evolution

To develop an understanding for how circulating deleterious mutations will impact patterns of influenza's antigenic evolution, we first extend existing population genetic models (*Haigh, 1978*; *Peck, 1994*) to acute infectious diseases undergoing immune escape. As is common in many population genetic models, we assume an infinite population and consider this population subject to frequent sublethal deleterious mutations that act independently from one another in reducing fitness. In an explicit susceptible-infected-recovered (SIR) epidemiological context that does not yet incorporate immune escape, these assumptions lead to a deleterious mutation-selection balance in the virus population ('Materials and methods', *Figure 1A*). This balance is given by a Poisson distribution with mean $\lambda/s_d$, where $\lambda$ is per-genome deleterious mutation rate and $s_d$ is the transmission fitness cost of sublethal deleterious mutations. The virus population at epidemiological equilibrium will have an overall net reproductive rate (i.e., mean absolute fitness) of $R = 1$, with more transmissible viruses that carry fewer deleterious mutations having reproductive rates above one and less transmissible viruses that carry more deleterious mutations having reproductive rates below one (*Figure 1A*, inset). This within-population variation in viral transmission rates is also reflected in the distribution of infected individuals' basic reproductive rates ($R_0$ values) (*Figure 1A* inset).

Following *Peck (1994)*, we first examine the fate of a single advantageous mutant arising in such a population. This advantageous mutant will necessarily arise in a genetic background with a certain number of deleterious mutations and, in our case, carry an immune-escape mutation that is beneficial to its spread. The genetic background in which the antigenic mutation arises and the size of the antigenic change jointly determine the mutant's initial reproductive rate $R_m(t = 0)$ in the virus population ('Materials and methods'). If this initial reproductive rate is less than one, the antigenic mutant is likely to be rapidly lost from the virus population. If the mutant's initial reproductive rate is instead greater than one, the mutant will invade the virus population if it is not initially stochastically lost. In the case of invasion, the mean reproductive rate of the antigenic mutant lineage will necessarily decline because it will accumulate its own set of deleterious mutations. (The mutant lineage's mean reproductive rate will also necessarily decline because the size of its susceptible host pool will decline over time, a factor we for now ignore but return to in later models.) For this invading antigenic mutant

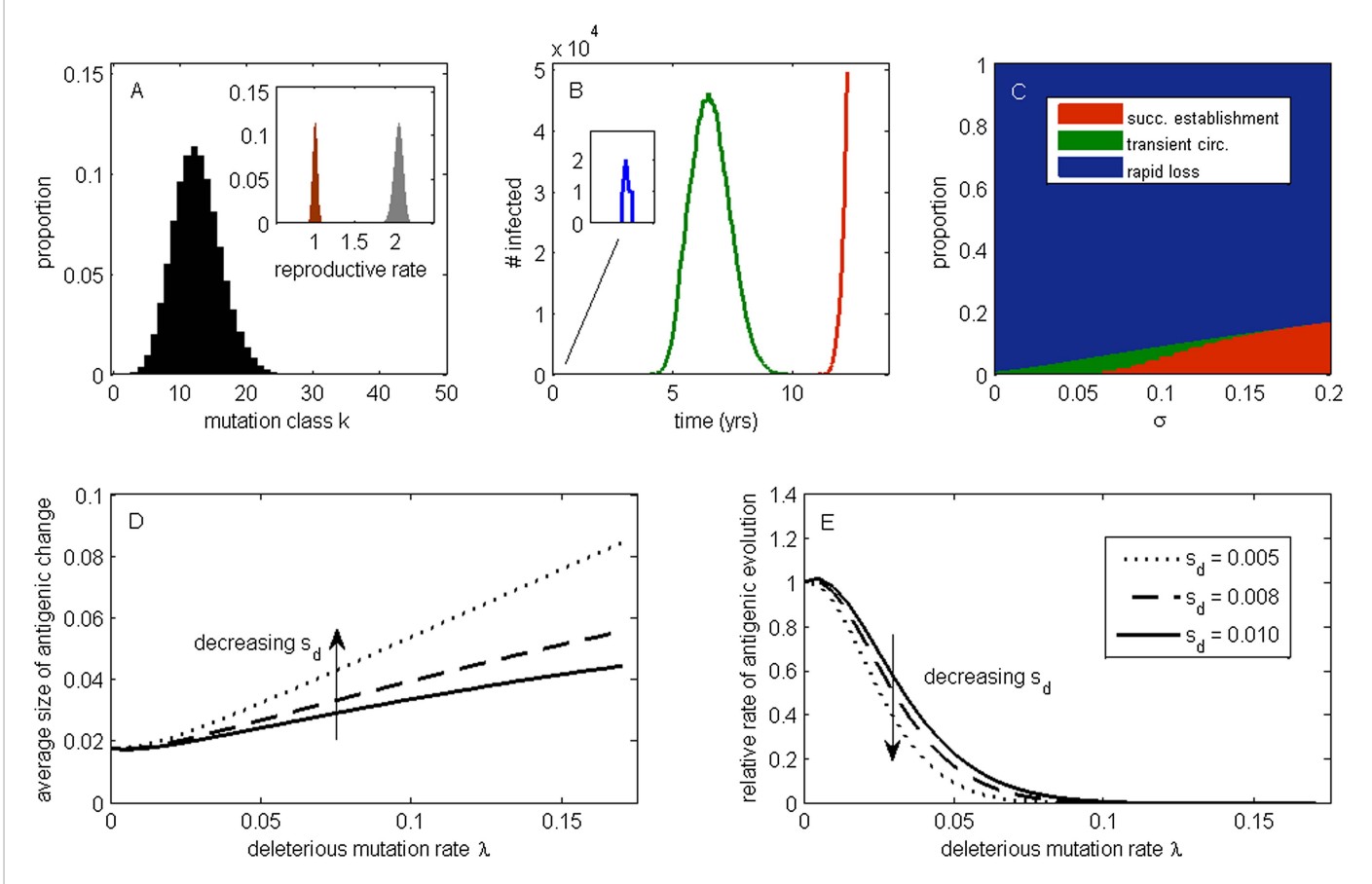

**Figure 1**. The effect of a deleterious mutation load on the fate of an antigenic mutant. (**A**) A resident antigenic strain at its deleterious mutation-selection balance. The histogram shows $p_k$, the proportion of infected individuals carrying a virus with $k$ deleterious mutations at its endemic equilibrium. The viral class carrying the fewest number of deleterious mutations is defined as mutation class $k = 0$. Inset: variation in the basic reproductive rate of infected individuals (gray histogram) and variation in the net reproductive rate $R$ of infected individuals (brown histogram) resulting from variation in the number of deleterious mutations carried by circulating viruses. Model parameters: $\lambda = 0.10$, $s_d = 0.008$, $\mu = 1/30$ years$^{-1}$, $\gamma = 1/4$ days$^{-1}$, $R_{0,k=0} = 2.25$. (**B**) Simulations showing the three dynamical fates that an antigenic mutant can experience: rapid loss (blue, with expanded inset), transient circulation (green), and successful establishment (red). The blue antigenic mutant (with $\sigma = 0.008$) arises in a genetic background with $k = 16$ deleterious mutations. The green antigenic mutant (with $\sigma = 0.04$) arises in a background of $k = 10$ deleterious mutations. The red antigenic mutant (with $\sigma = 0.06$) arises in a background of $k = 5$ deleterious mutations. All other parameters are as in (**A**). (**C**) The proportion of antigenic mutants that result in each of the three dynamical fates shown in (**B**) as a function of their antigenic size $\sigma$. All parameters as in (**A**). (**D**) The average antigenic size of successfully establishing antigenic mutants under different deleterious mutation rates $\lambda$ (x-axis) and for three transmission fitness costs (see (**E**) for legend). The size of arising antigenic mutations $\sigma$ are assumed to be gamma distributed with mean of 0.012 and a shape parameter of 2. All other parameters are as in (**A**). (**E**) The relative rate of antigenic evolution under different deleterious mutation rates $\lambda$ (x-axis) and for the three transmission fitness costs shown in (**D**). Other model parameters are as in (**D**). The relative rate of antigenic evolution is given by the fraction of arising antigenic mutants that establish under the deleterious mutation load relative to the fraction of arising antigenic mutants that would establish under a no-load scenario. See 'Materials and methods' for choice of model parameters.

lineage, we can calculate a final reproductive rate $R_m(t = \infty)$, which is the mean reproductive rate of this lineage once it has reached its own mutation-selection balance ('Materials and methods'). If this final reproductive rate falls below one, an invading antigenic mutant will therefore only transiently circulate before deterministically declining as a result of deleterious mutation accumulation. If this final reproductive rate exceeds one, however, the invading antigenic mutant lineage, under the assumptions of this model, will successfully establish. Antigenic mutants therefore experience one of three possible fates (*Figure 1B*): rapid loss, transient circulation, or successful establishment.

Which of these three fates awaits an antigenic mutant depends in part on the number of deleterious mutations carried by the strain in which the antigenic mutation arises: the lower the number of background deleterious mutations, the higher the antigenic mutant's chances are of at

least transient establishment, consistent with the background fitness interference effects found in *Illingworth and Mustonen (2012)*. Which fate occurs also depends on the extent to which the antigenic mutant escapes immunity (*Figure 1C*). The vast majority of small-sized antigenic mutants are rapidly lost; the remaining ones only circulate transiently before the accumulation of deleterious mutations results in their ultimate loss. Antigenic mutants that significantly escape immunity are less subject to rapid loss and also less likely to circulate only transiently. Given a specified size distribution for antigenic mutations, the average size of antigenic mutations that will successfully establish can be calculated. *Figure 1D* shows that this average antigenic size increases with increases in the deleterious mutation rate $\lambda$. The magnitude of the transmission fitness cost $s_d$ also affects the average size of antigenic mutants that will successfully establish. For any given deleterious mutation rate $\lambda$, as $s_d$ decreases the average size of antigenic mutants that fix increases (*Figure 1D*). These results can be interpreted in the context of findings from the population genetics literature: increases in $\lambda$ and decreases in $s_d$ similarly increase the virus population's fitness variance (as quantified by the variance in net reproductive rate $R$, *Figure 1A* inset, 'Materials and methods'). Increases in the fitness variance of asexual populations makes fixation of a beneficial mutant increasingly dependent on genetic background; only beneficial mutants that exceed a characteristic large size will have a high probability of fixing in populations with substantial fitness variance (*Peck, 1994*; *Barton, 1995*; *Schiffels et al., 2011*; *Good et al., 2012*).

In addition to their effect on the sizes of successfully establishing antigenic mutants, circulating deleterious mutations will act to slow the tempo of antigenic evolution (*Figure 1E*); that is, they will reduce the number of antigenic mutants that go to fixation in a given amount of time. This particular effect has previously been remarked upon in the context of a population genetics model for influenza's HA protein (*Strelkowa and Lässig, 2012*). Again, increases in the fitness variance of the viral population is the culprit: increases in the deleterious mutation rate $\lambda$ and decreases in the fitness cost of deleterious mutations $s_d$ similarly act to increase fitness variance; with increased fitness variance in the population, the genetic background in which an antigenic mutant needs to arise in to have a chance at fixation will be increasingly restrictive. This leads to a reduced tempo of antigenic change, consistent with a reduction in the rate of adaptation that is known from the population genetics literature (*Peck, 1994*; *Barton, 1995*).

Our model's findings can now be situated in the context of influenza A/H3N2's characterized evolutionary dynamics in humans. In particular, detailed antigenic analyses have demonstrated that this virus undergoes punctuated antigenic evolution, with predominantly single amino acid changes of large antigenic effect being responsible for the occurrence of antigenic cluster transitions (*Smith et al., 2004*; *Koel et al., 2013*). This dynamic is consistent with our results that antigenic evolution—as traced by lineages that ultimately fix—should occur via mutations of characteristically large size. These same analyses, as well as others (*Plotkin et al., 2002*), have further indicated that antigenic cluster transitions occur only every 2 to 6 years, an incredibly slow pace given the virus's high mutation rate and the need for only a single amino acid to substantially alter antigenicity. Again, this dynamic is consistent with our results that the tempo of antigenic evolution should be severely reduced by circulating deleterious mutations. Thus, our model posits that the punctuated and surprisingly slow nature of influenza A/H3N2's antigenic evolution are related features of this largely asexual, adapting population subject to a deleterious mutation load: adaptive evolution requires not only that large antigenic mutants occur, but also that they occur in good genetic backgrounds.

The above model contains a number of simplifying assumptions. Among these are that the susceptible host pool is negligibly affected over the time period of the antigenic mutant's establishment and that the successful establishment of an antigenic mutant will lead to the fixation of the mutant lineage in the virus population. Although these assumptions may be reasonable under some scenarios, they may not always be in an epidemiological context. For example, an antigenic mutant might invade sufficiently slowly to erode its frequency-dependent advantage prior to the exclusion of the existing antigenic lineage. Due to only partial cross-immunity between these lineages, this would lead to long-term coexistence of the variants. Another scenario is one of an antigenic mutant with a particularly large selective advantage: this mutant might burn through its susceptible host population so rapidly that its net reproductive rate drops significantly below one, leading to the possibility of its own extinction along with that of the previously circulating variant (*Ballesteros et al., 2009*). This scenario underscores the importance of considering the possibility of a variable virus population size; population genetic models that assume a constant population size may not be appropriate under certain epidemiological conditions.

To relax both the assumption of a time-invariant selective advantage and the assumption that successful establishment of a mutant leads to fixation of the mutant lineage (including replacement of the resident lineage), we now consider a more complex epidemiological model that explicitly incorporates the dynamics of susceptible hosts ('Materials and methods'). *Figure 2* shows the dynamics of the resident strain and the antigenic mutant under four distinct scenarios: a scenario in which a small antigenic mutant arises in a low-load ('good') genetic background (*Figure 2A*); a scenario in which a small antigenic mutant arises in an average-load genetic background (*Figure 2B*); a scenario in which a large antigenic mutant arises in a good genetic background (*Figure 2C*); and a scenario in which a large antigenic mutant arises in an average genetic background (*Figure 2D*). These simulations indicate that the three fates predicted in the simpler model still play out in the explicit context of epidemiological dynamics when parameterized for influenza A/H3N2 in humans. Rapid loss is expected to occur when a small antigenic mutant arises in an average genetic background (*Figure 2B*). As this combination occurs commonly, rapid loss is the most frequent fate experienced by antigenic mutants. Transient circulation occurs when a small antigenic mutant arises in a good genetic background (*Figure 2A*), provided that it survived genetic drift. Transient circulation also occurs when a large antigenic mutant arises in an average genetic background (*Figure 2D*), again provided that it survived genetic drift. Successful establishment can only occur when a large antigenic mutation arises in a good genetic background (*Figure 2C*), a 'jackpot' combination. In this case, the resident strain is competitively excluded as a result of strain cross-immunity. Intriguingly, the presence of deleterious mutations can also affect the invasion dynamics of antigenic mutants having this

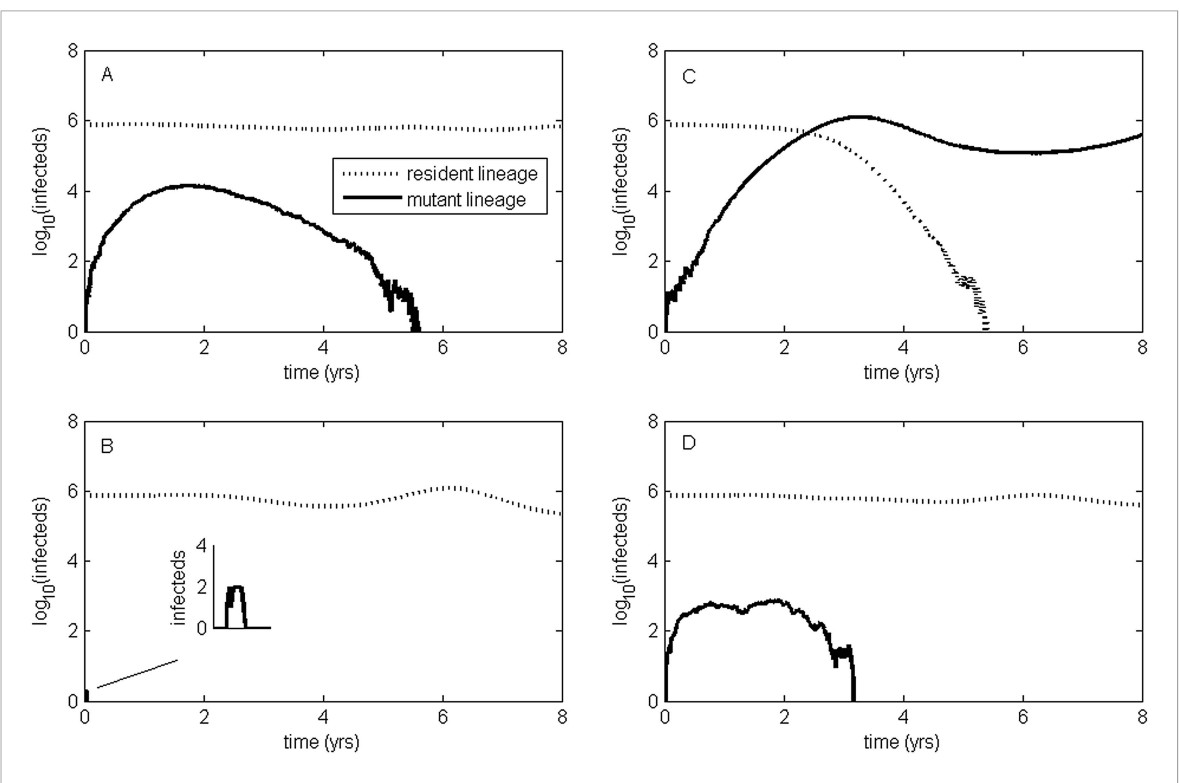

**Figure 2**. Epidemiological dynamics following the emergence of an antigenic mutant. First row: antigenic mutants arising in low-load genetic backgrounds ($k = 3$ deleterious mutations). Second row: antigenic mutants arising in average-load genetic backgrounds ($k = 13$ deleterious mutations). Left column: antigenic mutants that are of a small size ($\sigma = 0.004$). Right column: antigenic mutants that are of a large size ($\sigma = 0.045$). (**A**) Small antigenic mutants arising in good genetic backgrounds transiently circulate. (**B**) Small antigenic mutants arising in average genetic backgrounds are rapidly lost. (**C**) Large antigenic mutants arising in good genetic backgrounds can successfully establish and exclude the resident antigenic strain, resulting in an antigenic cluster transition. (**D**) Large antigenic mutants arising in average genetic backgrounds transiently circulate. All simulations assume a host population size of $N = 4$ billion and start with a single antigenic strain at evolutionary and epidemiological equilibrium. The remaining parameters are as in *Figure 1A*.

'jackpot' combination: because offspring of antigenic mutants accumulate deleterious mutations, and these deleterious mutations reduce viral transmissibility, the invasion dynamics of particularly large antigenic mutants are considerably less explosive than would be expected in the absence of deleterious mutation accumulation (*Figure 3*). Consequently, large antigenic mutants do not readily burn themselves out during attempted establishment, as might in theory be expected (*Ballesteros et al., 2009*).

## Antigenic cluster transitions can occur via three distinct molecular pathways

*Figure 2C* shows that an antigenic mutant can exclude a resident antigenic strain, provided that it arises in a good genetic background and carries an antigenic mutation of large effect. This is consistent with detailed molecular studies of influenza A/H3N2 that have shown that single amino acid changes of large antigenic effect can precipitate an antigenic cluster transition (*Smith et al., 2004*; *Koel et al., 2013*), although the importance of the genetic background in which the antigenic mutation arises has not been discussed in the context of this work. The BE92-to-WU95 cluster transition, precipitated by an amino acid change from N to K at site 145, is a good example of a cluster transition occurring via a single mutational step (*Koel et al., 2013*) (*Figure 4A*). Of note,

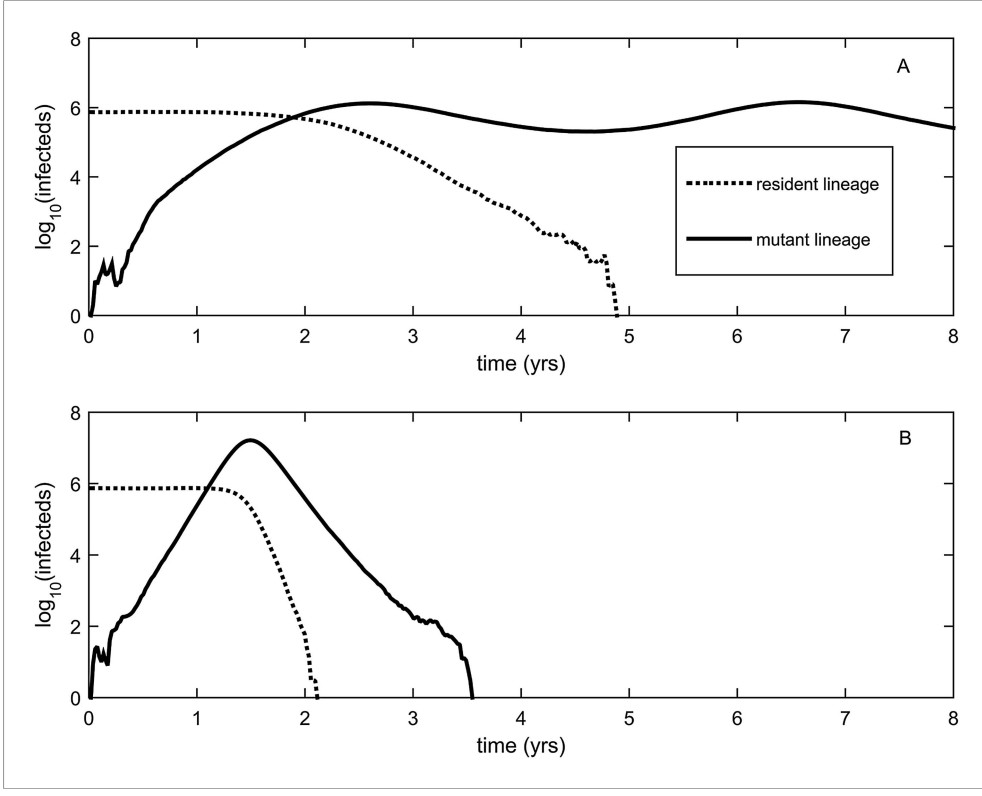

**Figure 3**. Explosiveness in cluster invasion dynamics in the presence and absence of deleterious mutation accumulation. (**A**) A representative example of the population dynamics of an antigenic mutant in the presence of deleterious mutation accumulation. The new antigenic strain invades and successfully establishes, while excluding the resident strain, characteristic of a successful antigenic cluster transition. Model parameters are $N = 4$ billion, $\mu = 1/30$ years$^{-1}$, $\gamma = 1/4$ days$^{-1}$, $\lambda = 0.10$, $s_d = 0.008$, $R_{0,k = 0} = 2.25$, and $\sigma = 0.05$. In this simulation, the antigenic mutant arises in a genetic background with $k = 4$ deleterious mutations. (**B**) A representative example of the population dynamics of an antigenic mutant in the absence of deleterious mutations. The new antigenic strain invades explosively, leading to its own burn-out along with exclusion of the resident strain. Model parameters are $N = 4$ billion, $\mu = 1/30$ years$^{-1}$, $\gamma = 1/4$ days$^{-1}$, $R_0 = 2.04$, and $\sigma = 0.121$. The value of $R_0$ was chosen such that, prior to the invasion of the antigenic mutant, the fraction of the host population susceptible to infection and the number of infected hosts was the same across the two simulations. In (**B**), the value of $\sigma$ was chosen such that $R_m(t = 0)$ was the same across the two simulations (at a value of 1.13).

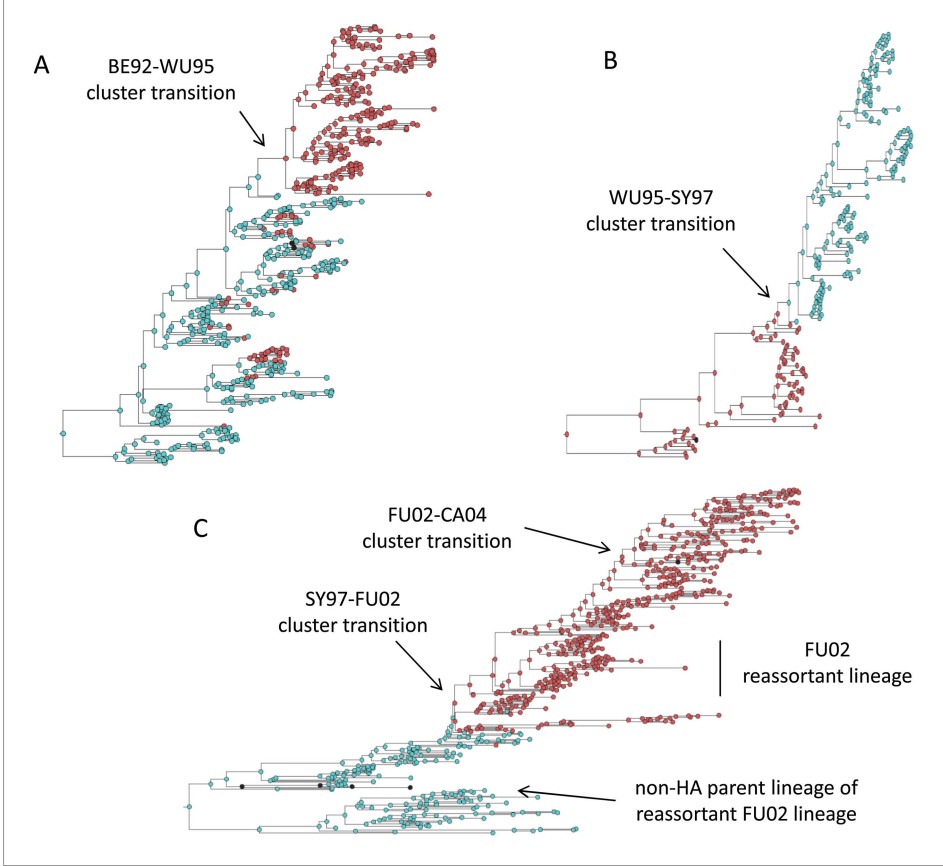

**Figure 4**. Influenza phylogenies consistent with the three distinct molecular pathways by which antigenic cluster transitions may occur. (**A**) Maximum clade credibility (MCC) phylogeny showing the BE92-WU95 antigenic cluster transition, reconstructed from sequences spanning years 1993–1997. The phylogeny shows evolutionary dynamics that are consistent with a 'jackpot' combination of a large antigenic mutation arising in a rare low-load genetic background. Hemagglutination inhibition assays experimentally indicated that only a single amino acid change of large antigenic effect (145NK) was necessary to precipitate the cluster transition (*Koel et al., 2013*). Nodes are colored by the amino acid present at this site (145N = blue; 145K = red; other = black). (**B**) MCC phylogeny showing the WU95-SY97 antigenic cluster transition, reconstructed from sequences spanning years 1995–1999. The phylogeny shows evolutionary dynamics consistent with the two-step antigenic change molecular pathway leading to antigenic cluster transitions, as depicted in *Figure 5A*. Hemagglutination inhibition assays experimentally indicated that two amino acid changes (156KQ and 158EK) were necessary to precipitate the cluster transition (*Koel et al., 2013*). Nodes are colored by the amino acids present at these sites (156K158E = red; 156Q158K = blue; other = black). Note that both amino acid changes occur on the same short internal branch, such that this apparently rapid transition is unlikely to be an artifact of sparse sampling. (**C**) MCC phylogeny showing the SY97-FU02 and FU02-CA04 cluster transitions, reconstructed from sequences spanning years 2001–2005. The phylogeny shows evolutionary dynamics consistent with the two-step reassortant molecular pathway leading to antigenic cluster transitions, as depicted in *Figure 5B*. Hemagglutination inhibition assays experimentally indicated that only a single amino acid change (156QH) antigenically defined the SY97-FU02 cluster transition (*Koel et al., 2013*). Nodes are colored by the amino acid present at this site (156Q = blue; 156H = red). Vertical bar shows the FU02 reassortant clade. The genetically distant non-hemagglutinin (HA) parent lineage of this reassortant clade is also shown. Phylogenies in (**A–C**) were inferred using BEAST (*Drummond et al., 2012*) from full HA1 sequences with specified sampling dates.

several clades witnessed the 145NK amino acid substitution before it occurred in the clade that founded the WU95 viral lineage. Our models above indicate that the failure of these early 145K clades to establish could in principle be a consequence of this N-to-K amino acid change occurring in insufficiently good genetic backgrounds, as also suggested in *Neher et al. (2014)*. Other explanations for the failure of early 145K clades to establish are of course possible. One such explanation is that

these clades might have circulated in spatial locations that are not sufficiently well-connected globally. Recent work has further emphasized the important role that spatial ecology plays in the global establishment of antigenic variants (*Russell et al., 2008*; *Lemey et al., 2014*; *Bedford et al., 2015*), with findings suggesting that Asia plays a dominant role in sourcing these new variants. Thus, if the early 145K clades were not geographically well-situated, the spatial context (rather than the genetic context) of these clades may have led to their failure in establishing. We thus examined the spatial locations of the sequences from the three largest 145K clades that failed to establish: all three clades were geographically widespread, spanning at least two continents. Furthermore, two of the three clades contained sequences from Asia, despite Asia being undersampled during this time period. It is thus unlikely that these early 145K clades were geographically restricted to 'sink' populations. An alternative explanation for the failure of these early 145K clades to establish is that herd immunity levels against BE92 may not yet have been high enough to result in a sufficiently large selective advantage for these WU95-like lineages. Given these alternative explanations, an in vitro experimental study that quantifies relative viral fitness of these 145K lineages would thus be necessary to confirm our genetic background hypothesis.

While hitting the 'jackpot' combination is a viable molecular pathway for precipitating an antigenic cluster transition, the combination of a large antigenic mutation arising in a low-load genetic background is expected to occur infrequently. We can therefore consider whether there might be alternative molecular pathways open to antigenic mutants that would similarly yield a successful cluster transition. One possibility is for a more common small- to medium-sized antigenic mutation to first occur in a good genetic background. This would result in a transient rise of this mutant (*Figure 2A*), thereby increasing the number of individuals infected with the virus carrying few deleterious mutations. A less common, large-sized antigenic mutation could then occur, effectively piggy-backing on the good genetic background that the smaller antigenic mutant inflated. Because smaller antigenic mutations can only circulate transiently, and accumulate deleterious mutations during their circulation, the large antigenic mutation must not only follow, but also rapidly follow, the rise of the smaller antigenic mutation for this molecular pathway to yield an antigenic cluster transition. This scenario is depicted in *Figure 5A*, and phylogenetically would result in a sudden appearance of a viral lineage carrying two antigenic mutations. A phylogenetic analysis of the WU95-SY97 antigenic cluster transition provides an empirical example that is consistent with this molecular pathway of antigenic turnover, with a seemingly simultaneous accumulation of two antigenic amino acid changes (156KQ and 158EK) occurring on a short branch of the reconstructed phylogeny (*Koel et al., 2013*) (*Figure 4B*).

A third molecular pathway that could in principle precipitate an antigenic cluster transition is for a large antigenic mutation to first arise in an average genetic background and, during its transient circulation (*Figure 2D*), to purge itself of a large number of deleterious mutations. This purging could arise through within-subtype viral reassortment taking place in an individual coinfected with a strain belonging to the resident cluster and a strain belonging to the transiently circulating antigenic mutant. Even though both of the strains infecting this individual would likely carry an average deleterious mutation load, it is highly unlikely that they will carry the same set of deleterious mutations if phylogenetically sufficiently far apart. Reassortment of the eight gene segments within the coinfected host could therefore significantly lower the number of deleterious mutations carried by viral progeny characterized as belonging to the new antigenic cluster. Once the deleterious mutation load has largely been shed, the reassortant virus would quickly rise and cause an antigenic cluster transition (*Figure 5B*). Indeed, many historical instances of intrasubtypic reassortment contributing to antigenic turnover have been documented (*Morens et al., 2009*); these instances have been associated with high incidence levels as would be expected by purging of deleterious mutations. A phylogenetic analysis of the SY97-FU02 cluster transition provides an especially compelling example that is consistent with this molecular pathway of antigenic turnover. In this cluster transition, a virus antigenically characterized as FU02 reassorted with a genetically distant virus antigenically characterized as SY97 (*Barr et al., 2005*; *Holmes et al., 2005*) (*Figure 4C*). This reassortant viral lineage circulated extensively in Australia and New Zealand in 2003 and subsequently in the US in the 2003–2004 influenza season, causing substantial morbidity and mortality (*Barr et al., 2005*). Although this viral lineage may ultimately have led to the replacement of the non-reassortant FU02 viral lineage, both of these FU02 lineages were excluded by the subsequent CA04 antigenic cluster, which appears to have originated from the non-reassortant FU02 lineage (*Figure 4C*). Intriguingly, the CA04 lineage

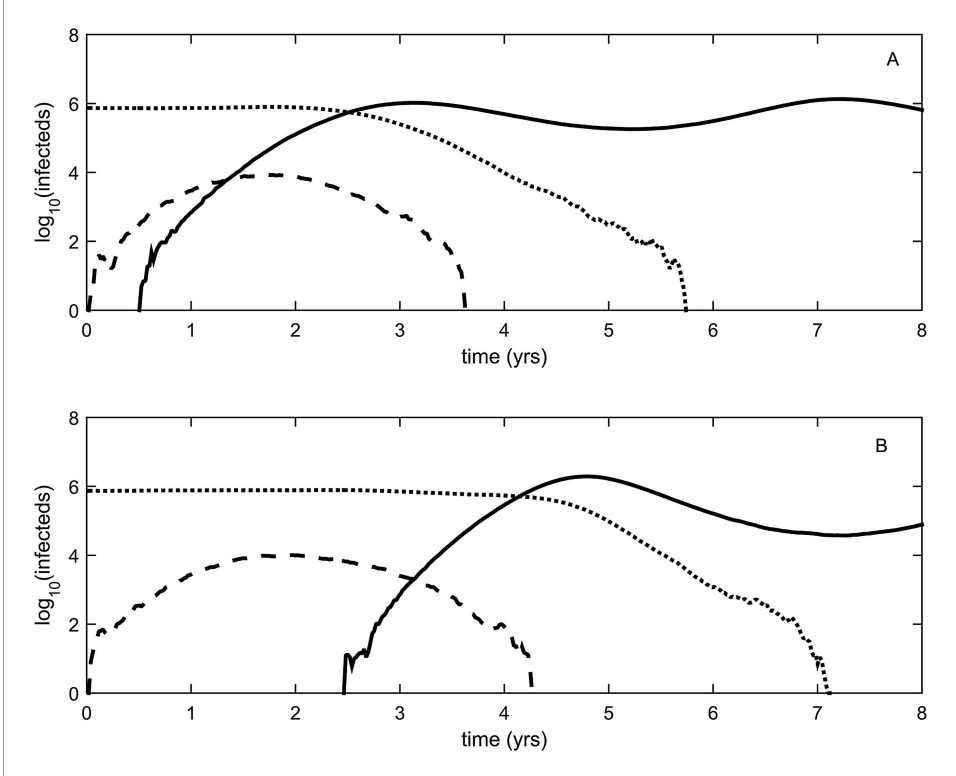

**Figure 5**. Two-step approaches to antigenic cluster transitions. (**A**) A cluster transition arising from two consecutive antigenic mutations. A small antigenic mutation ($\sigma = 0.003$, dashed line) first arises in a good genetic background (deleterious mutation load $k = 2$) of the resident strain (dotted line). Shortly after, a second and larger-sized antigenic mutation ($\sigma = 0.045$, solid line) occurs in an individual infected with the single antigenic mutant. This sequence of events precipitates an antigenic cluster transition, with the double mutant replacing the resident strain and the low-frequency single mutant. We assume that the degree of immune escape is additive, such that $\sigma$ between the resident strain and the double mutant is $\sigma = 0.048$. (**B**) A cluster transition arising from intrasubtypic viral reassortment. A large-sized antigenic mutation ($\sigma = 0.06$, dashed line) first arises in an average genetic background ($k = 10$) of the resident strain (dotted line). After 2.5 years of circulation, a coinfection that leads to the generation of low-load mutant ($k = 4$) occurs. This low-load mutant (solid line) replaces the resident strain and the average-load carrying antigenic mutant, ultimately precipitating an antigenic cluster transition. Other model parameters in (**A**) and (**B**) are $N = 4$ billion, $\mu = 1/30$ years$^{-1}$, $\gamma = 1/4$ days$^{-1}$, $R_{0,k=0} = 2.25$, $\lambda = 0.10$, and $s_{d} = 0.008$.

carried with it not only an HA mutation of large antigenic size, but also two amino acid changes in its polymerase gene segment that enhanced replicative fitness (*Memoli et al., 2009*).

## Antigenic evolution in the context of fitness variation generated by deleterious mutations results in a spindly HA phylogeny, antigenic variant co-circulation, and high annual attack rates

Our above analyses have relied on simple epidemiological models to gain intuition for how circulating sublethal deleterious mutations would impact patterns of influenza A/H3N2's antigenic evolution. These analyses indicate that deleterious mutation loads should lower the rate of antigenic evolution (*Figure 1E*). Based on previous modeling work (*Koelle et al., 2006*, *2009*, *2010*; *Zinder et al., 2013*), a lower rate of antigenic evolution is known to constrain genetic and antigenic diversity and thus might lead to a spindly HA phylogeny. Our analyses also indicate that deleterious mutation loads increase the average size of antigenic variants that establish in the long run (*Figure 1D*); observed patterns of punctuated antigenic evolution (*Smith et al., 2004*) may thus be better reproduced with a model that integrates sublethal deleterious mutations than one that ignores these mutations. Furthermore, our above analyses indicate that antigenic variants can reach appreciable numbers even

when their ultimate fate is one of only transient circulation. This pattern of transient circulation points towards the possibility of co-circulation of a substantial number of antigenic variants, as argued for in statistical analyses of influenza's HA sequences (*Strelkowa and Lässig, 2012*). In our model, these antigenic variants need not necessarily strongly compete with one another for susceptible hosts for only a single lineage to persist. In the absence of exceptionally strong competition for susceptible hosts, the co-circulation of these antigenically distinct variants may thus be capable of reproducing empirically observed high annual attack rates (*World Health Organization, 2014*).

To determine whether spindly HA phylogenies, co-circulation of antigenic variants, and high annual attack rates indeed come out of a model that incorporates deleterious mutations, we implemented a phylodynamic model that simulates the occurrence of both non-antigenic (largely deleterious) mutations and antigenic mutations ('Materials and methods'). When simulated under parameters appropriate for influenza A/H3N2 in humans, this model yields a spindly HA phylogeny with low-load viruses populating the trunk and higher load viruses populating the tips of the phylogeny (*Figure 6A*).

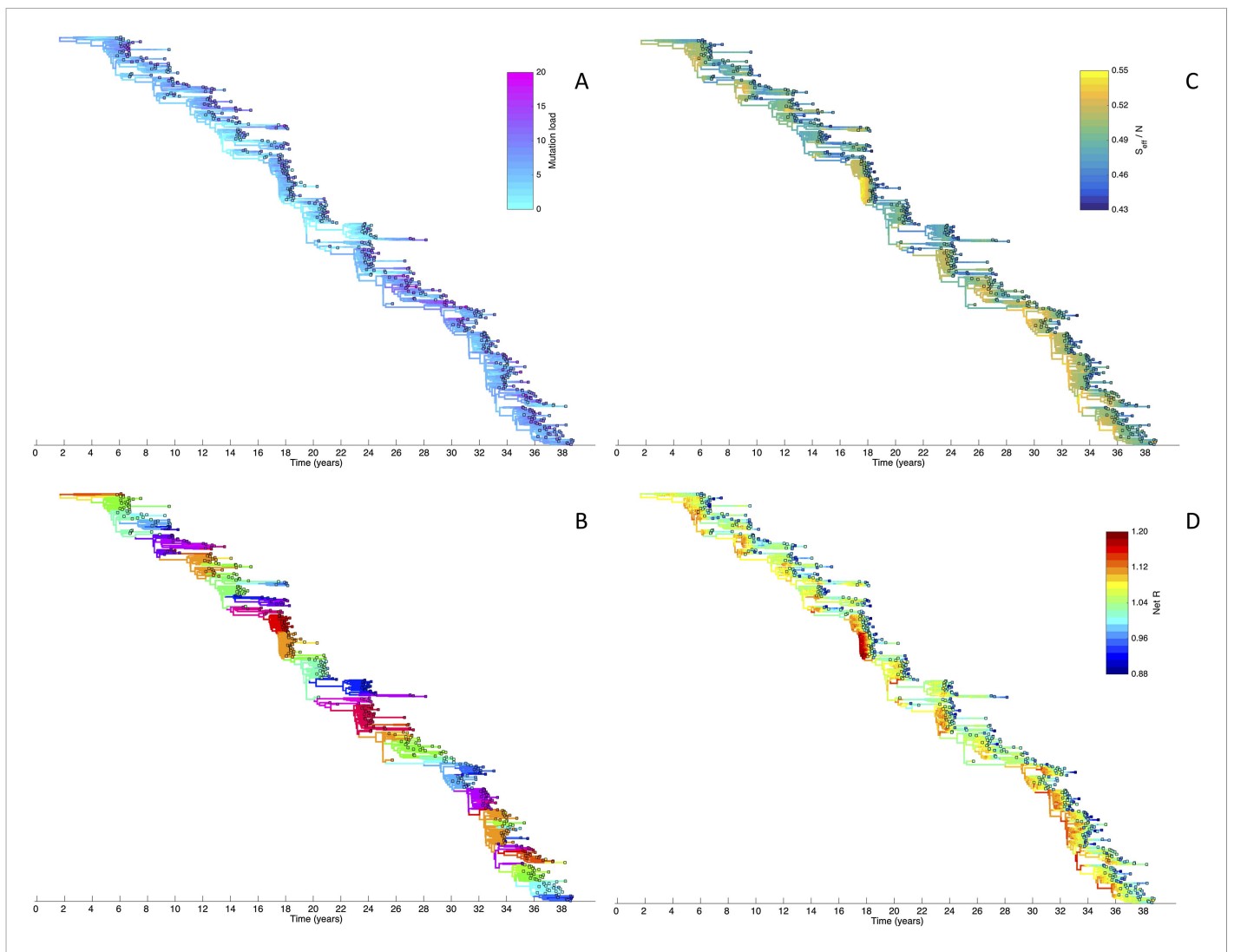

**Figure 6**. Viral phylogenies from a simulation of the phylodynamic model incorporating antigenic and non-antigenic mutations. (**A**) Simulated phylogeny reproducing H3N2's spindly HA phylogeny, with low levels of genetic and antigenic diversity over the long run. Lineages are colored according to their deleterious mutation loads. (**B**) Simulated phylogeny shown in (**A**) with lineages colored according to their antigenic type. Similarly colored lineages that are genetically distinct are antigenically distinct (colors were re-used due to their limited number). (**C**) Simulated phylogeny shown in (**A**) with lineages colored according to the fraction of the host population susceptible to infection with that lineage ($S_{eff}/N$). (**D**) Simulated phylogeny shown in (**A**) with lineages colored according to their net reproductive rate $R$.

This distribution of deleterious mutation loads on the simulated phylogeny is consistent with the excess number of non-synonymous substitutions on external tree branches that was found for human influenza A/H3N2 (*Fitch et al., 1997*; *Pybus et al., 2007*) and arises because deleterious mutations contribute a substantial fraction of the fitness variance in the viral population (*Figure 7A*).

The phylodynamic model simulation further reproduces antigenic variant co-circulation (*Figures 6B, 7B*), consistent with findings that lineages that are lost nevertheless undergo appreciable antigenic evolution (*Strelkowa and Lässig, 2012*) and that antigenic diversity levels within clusters can exceed antigenic distances between clusters (*Smith et al., 2004*). The simulation yields prevalence levels of 10–180 cases per 100,000 individuals (*Figure 7B*) and annual attack rates of approximately 2–10%, consistent with empirical estimates of influenza incidence (*World Health Organization, 2014*).

Despite extensive co-circulation of antigenic variants, the overall phylogeny remains ladder-like due to the selective sweeps initiated by rare large-sized antigenic mutations that arise in good genetic backgrounds. This dynamic is evident by jointly considering *Figure 6A* and *Figure 6C*, with *Figure 6C* showing, for each lineage, the fraction of the host population that is susceptible to infection with that lineage. From these figures, it is clear that the trunk of the phylogeny carries low-load viruses (*Figure 6A*) that have a high number of susceptible hosts (*Figure 6C*). This combination together yields high-fitness viruses, as epidemiologically given by their net reproductive rates $R$ (*Figure 6D*). It is these viruses that establish and thus form the trunk of the tree. Neither a low mutation load alone nor a high number of susceptible hosts alone suffice in generating a sufficiently high-fitness viral lineage that will ensure its long-term evolutionary success.

Inspection of *Figure 6C* also shows that trunk lineages abruptly gain susceptible hosts. These abrupt gains are a result of single large antigenic mutations that, when occurring in good genetic backgrounds, initiate the selective sweeps that ultimately limit influenza's genetic diversity. How these selective sweeps affect levels of standing genetic diversity in the viral population is shown in *Figure 7C*, where we use the time to the most recent common ancestor (tMRCA) of all circulating lineages as a proxy for total genetic diversity. This tMRCA plot reproduces quantitative features of influenza A/H3N2's tMRCA plot presented in *Bedford et al. (2011)*, including its interannual variation and the observed major drops in tMRCA following the emergence of new and antigenically very distinct clusters.

The importance of deleterious mutations in constraining the genetic and antigenic diversity of influenza can be further illustrated by simulating the phylodynamic model under the assumption of their absence. These simulations very rapidly yield explosive genetic and antigenic diversity (*Figure 8A*) and, as a consequence, prevalence levels that generally increase over time (*Figure 8B*). Reassuringly, this finding is consistent with predictions from previous influenza modeling work incorporating only strain-specific immunity (*Ferguson et al., 2003*).

Given that purifying selection alone is known to reduce genetic diversity (*Charlesworth et al., 1993*; *Walczak et al., 2012*), we further simulated the phylodynamic model under the assumption of no antigenic mutations. In these simulations, we phenomenologically incorporated antigenic drift by simulating susceptible-infected-recovered-susceptible (SIRS) dynamics such that prevalence levels were similar to those shown in *Figure 7B*. Compared with the results shown in *Figure 6*, these simulations gave rise to significantly higher levels of genetic diversity (*Figure 9A*) and longer tMRCAs (*Figure 9B*). This indicates that purifying selection alone does not account for the spindly phylogenies shown in *Figure 6*. Rather, it is antigenic evolution in the context of fitness variation generated by deleterious mutations that constrains the viral phylogeny.

While the above simulations demonstrate that a model incorporating both antigenic and deleterious mutations can reproduce influenza A/H3N2's spindly phylogeny, its antigenic co-circulation patterns, and its high annual attack rates, a number of the parameters that required specification are empirically not well characterized. Most notably, these are the evolutionary parameters of the model: the deleterious mutation rate $\lambda$, the fitness cost of deleterious mutations $s_d$, and the antigenic mutation rate $\lambda_{antigenic}$. In *Figure 10*, we show how the evolutionary and epidemiological dynamics of the model simulations depend on these three parameters. Specifically, we vary one parameter while keeping the remaining two parameters fixed. For each parameter set considered, we perform 20 simulations to determine the range of dynamics predicted under a single parameterization. For each simulation, we quantify the virus's evolutionary dynamics by plotting the minimum and maximum tMRCA calculated over a continuous 10-year period (years 15–25 of the

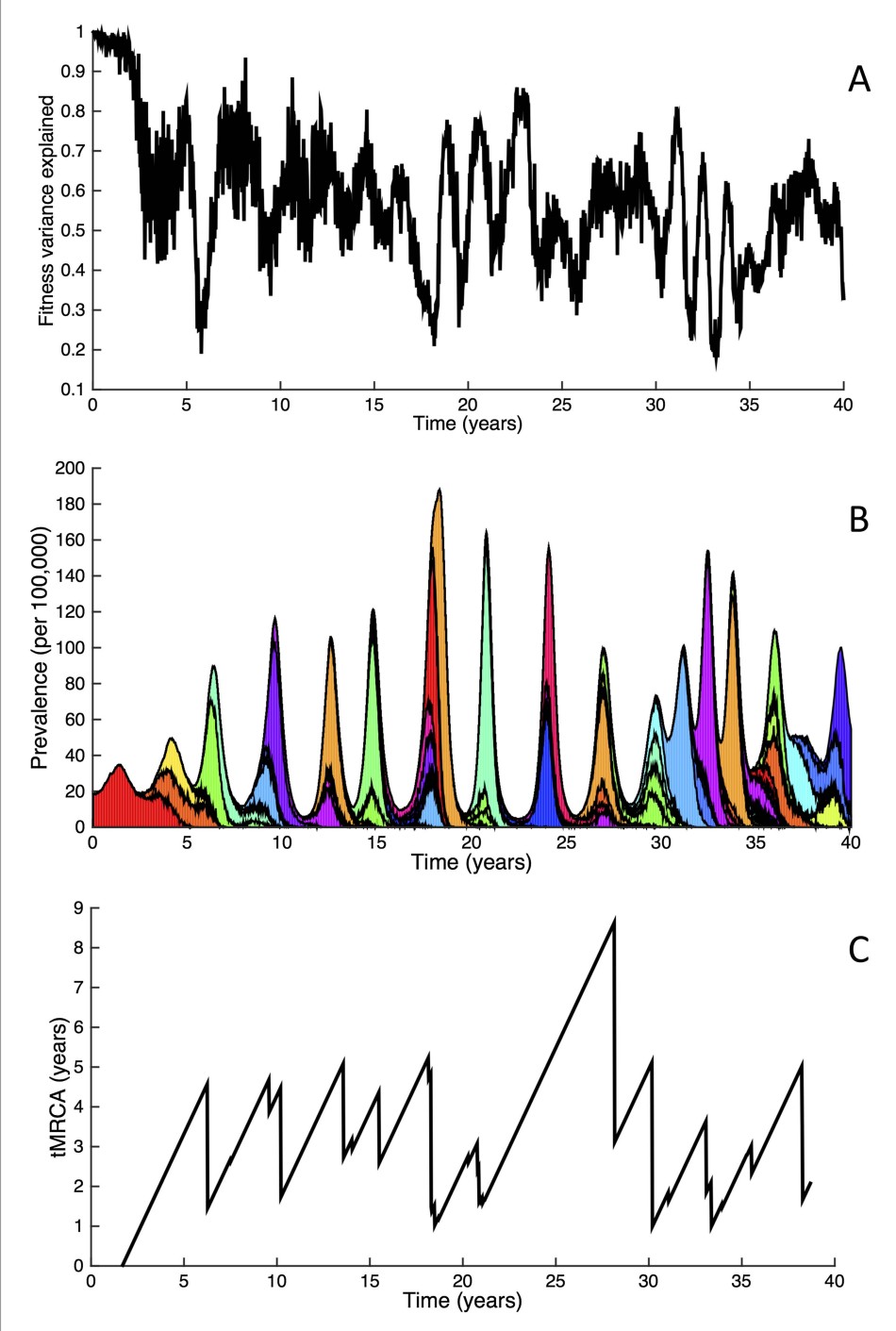

**Figure 7**. Fitness variance dynamics, epidemiological dynamics, and times to most recent common ancestor for a simulation of the phylodynamic model incorporating antigenic and non-antigenic mutations. (**A**) The fraction of (log) viral fitness variation explained by deleterious mutations over time for the simulation whose phylogenies are plotted in *Figure 6*. The fraction not explained by mutation load is due to antigenic variation in the population. (**B**) Simulated epidemiological dynamics showing co-circulation of multiple antigenic variants and sustained prevalence levels over time for this same simulation. (**C**) Times to the most recent common ancestor (tMRCAs), computed over time from the phylogenies shown in *Figure 6*.

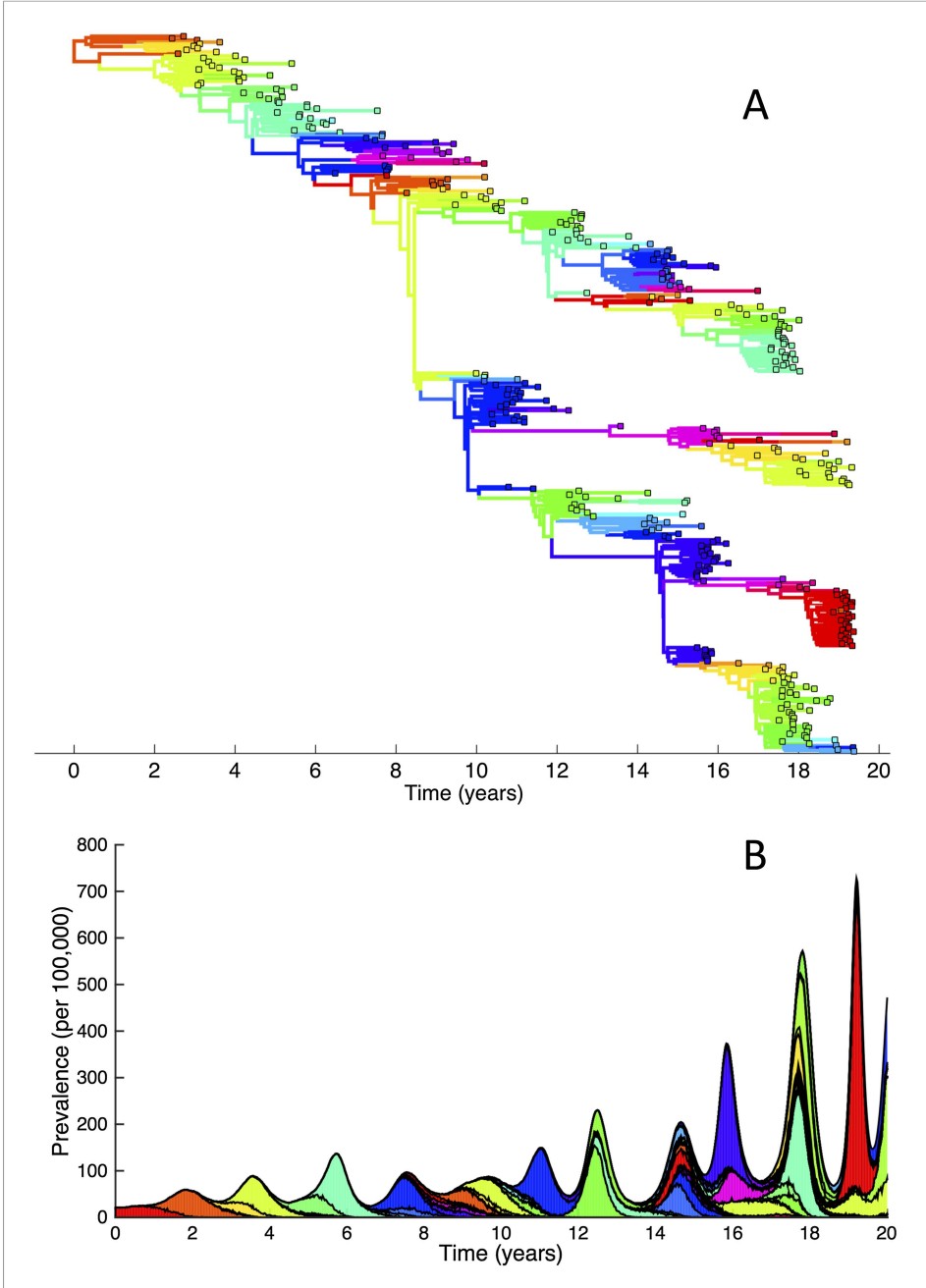

**Figure 8**. Simulations of the phylodynamic model in the absence of a deleterious mutation load. (**A**) Simulated phylogeny showing explosive genetic and antigenic diversity over a 20-year period. Lineages are colored according to their antigenic type, with similarly colored lineages that are genetically distinct being antigenically distinct (colors were re-used due to their limited number). (**B**) Simulated epidemiological dynamics showing prevalence levels generally increasing over time.

simulation) as a measurement of the extent of genetic diversity in the viral population. To quantify the virus's epidemiological dynamics, we plot the minimum and maximum annual attack rates over this same time period. *Figure 10A* shows that in the absence of deleterious mutations ($\lambda = 0$) the model simulations yield explosive viral diversity, with the maximum tMRCA ranging between 5 and 25 years. The observed increases in viral genetic and antigenic diversity result in unrealistically high maximum annual attack rates, in the range of 15–45% (*Figure 10B*). Simulated under this parameterization, the

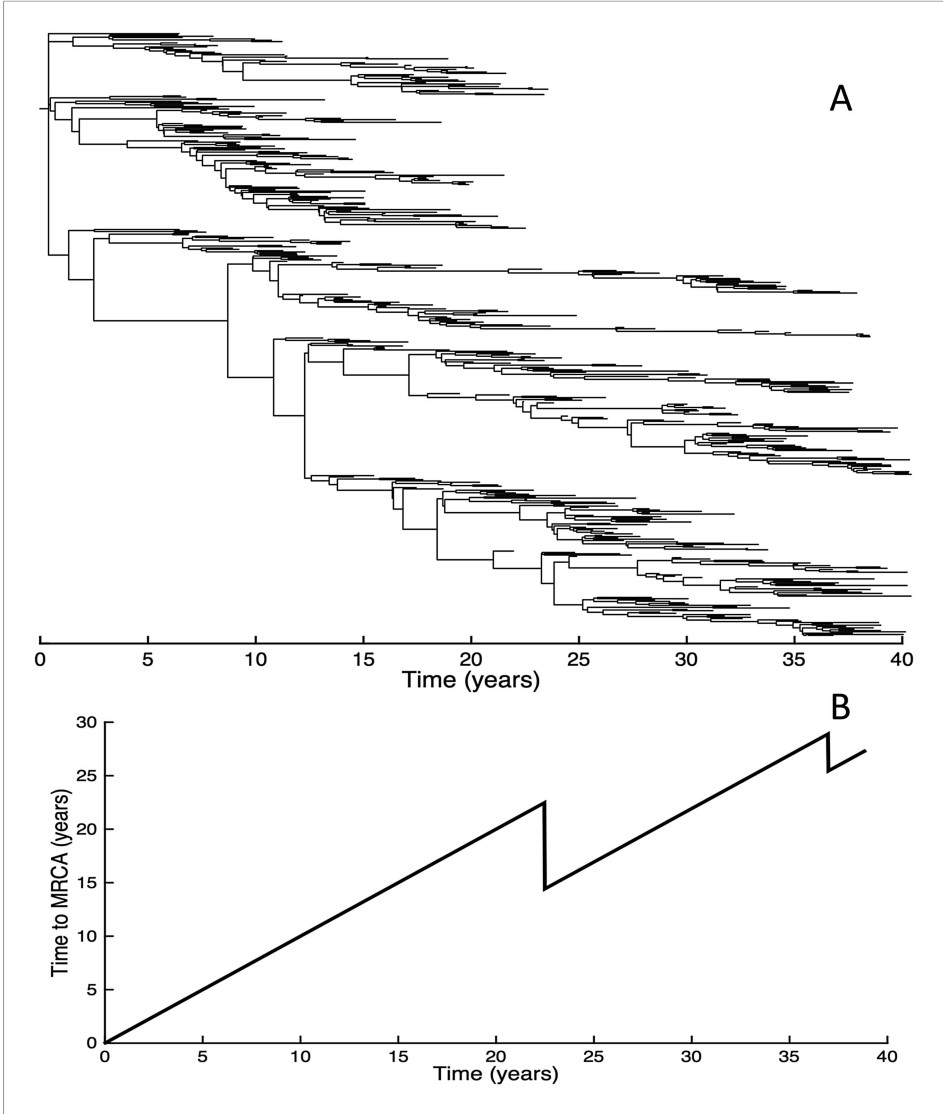

**Figure 9**. Simulations of the phylodynamic model in the absence of antigenic mutations. (**A**) Simulated phylogeny under a parameterization with no antigenic mutations, showing genetic diversity generally increasing over time. (**B**) Times to the most recent common ancestor (tMRCAs) computed over time from the phylogeny shown in (**A**).

results shown in **Figure 8A,B** are representative of these results. With increasing deleterious mutation rates, the maximum tMRCAs decline as do maximum annual attack rates (**Figure 10A,B**). For $\lambda$ values of 0.10 and higher, empirically documented annual attack rates can be consistently reproduced. Maximum and minimum tMRCAs are best reproduced for $\lambda$ values between 0.10 and 0.15.

**Figure 10C,D** shows the evolutionary and epidemiological effects of the fitness cost of deleterious mutations, $s_d$. It is clear from these plots that neither the mean maximum nor the mean minimum tMRCA across the simulations depends strongly on $s_d$ (**Figure 10C**). However, the range of maximum tMRCA values is considerably higher at lower $s_d$ values (**Figure 10C**). This may be because selective sweeps are expected to occur more rarely at lower $s_d$ values (**Figure 1E**), such that the viral population is homogenized less frequently at these lower values, leading to higher tMRCA values. This explanation is consistent with the slightly lower annual attack rates at lower $s_d$ values (**Figure 10D**): when the rate of antigenic evolution is slower, individuals cannot be reinfected as rapidly and thus annual attack rates would be lower. Despite the dependency of evolutionary and epidemiological dynamics on $s_d$, **Figure 10C,D** shows that a broad range of $s_d$ values yields dynamics that are consistent with influenza A/H3N2 dynamics in humans.

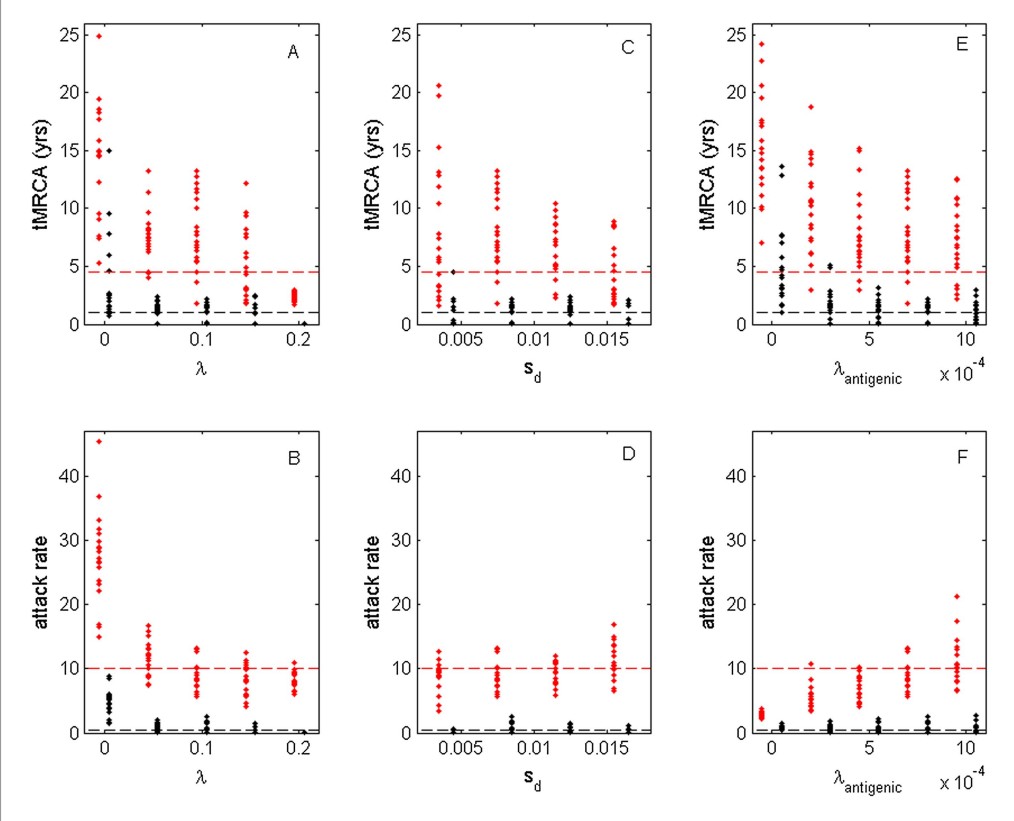

**Figure 10**. Sensitivity of evolutionary and epidemiological dynamics to parameters of the phylodynamic model. Subplots (**A**, **B**) show model sensitivity to the deleterious mutation rate $\lambda$. Subplots (**C**, **D**) show model sensitivity to the fitness cost of deleterious mutations $s_d$. Subplots (**E**, **F**) show model sensitivity to the antigenic mutation rate $\lambda_{antigenic}$. The top row shows maximum (red dots) and minimum (black dots) times to the most recent common ancestor (tMRCAs) for 20 independent simulations. The red dashed line indicates the maximum tMRCA inferred from a phylogenetic analysis of influenza A/H3N2's HA (**Bedford et al., 2011**); the black dashed line indicates the minimum tMRCA inferred from this same analysis. The bottom row shows maximum (red dots) and minimum (black dots) annual attack rates for the same 20 simulations. The red dashed line indicates an estimate of the maximum annual attack rate for influenza A/H3N2; the black dashed line indicates an estimate of the minimum annual attack rate for influenza A/H3N2. These values are based on annual attack rate estimates in adults of 5–10%, such that the maximum annual attack rate is on the order of 10%, and the minimum annual attack rate is shown at 1% (which would correspond to years of negligible circulation of this influenza subtype). Each simulation was run for 28 years, and minimum and maximum tMRCAs and attack rates were computed from years 15–25 of the simulation. In subplots (**A**) and (**B**), $\lambda$ is varied, $s_d = 0.008$ and $\lambda_{antigenic} = 0.00075$. In subplots (**C**) and (**D**), $\lambda = 0.10$, $s_d$ is varied, and $\lambda_{antigenic} = 0.00075$. In subplots (**E**) and (**F**), $\lambda = 0.10$, $s_d = 0.008$, and $\lambda_{antigenic}$ is varied. All other parameter values are as listed in *Figure 6*.

Finally, *Figure 10E,F* shows the sensitivity of the model to the antigenic mutation rate $\lambda_{antigenic}$. In the absence of antigenic evolution ($\lambda_{antigenic} = 0$), maximum tMRCAs are significantly higher than empirically documented (*Figure 10E*) and maximum annual attack rates do not exceed 3.8% (*Figure 10F*). These findings are consistent with, *Figure 9*, which indicates that a spindly viral phylogeny cannot be reproduced under purifying selection alone in a model that is parameterized for influenza. (*Figure 9*'s model differs slightly from the model parameterized with $\lambda_{antigenic} = 0$, whose simulations are shown in *Figure 10E,F*: *Figure 9* shows simulation results from an SIRS model that yields annual attack rates consistent with those empirically observed for flu; the results shown in *Figure 10E,F* under $\lambda_{antigenic} = 0$ use the same parameterization as in *Figure 6*, with the exception of $\lambda_{antigenic}$, and thus simulate a simple SIR model. In either case, purifying selection alone cannot reproduce a spindly phylogeny.) At increasing $\lambda_{antigenic}$ values, tMRCAs decrease and annual attack

rates increase to empirically documented values. These plots further indicate that influenza's evolutionary and epidemiological dynamics can be reproduced over a wide range of antigenic mutation rates as long as the rate exceeds a certain minimum value. Finally, taken together, *Figure 10A,B,E,F* again indicates that it is the interaction between deleterious and advantageous immune escape mutations that consistently yields a spindly phylogeny and high annual attack rates. Neither deleterious mutations nor immune escape mutations alone succeed in reproducing these dynamic features of influenza.

## Discussion

Here, we have shown that population genetic and population dynamic models incorporating sublethal deleterious mutations can reproduce the characteristic features of influenza A/H3N2's evolutionary dynamics in humans. These include the virus's rare punctuated antigenic evolution and the low genetic diversity of its hemagglutinin protein. The low genetic diversity of the virus's HA, reflected in the spindliness of its phylogeny, has been a particular evolutionary characteristic that influenza modelers have sought to reproduce. To date, three other models exist that can explain this evolutionary characteristic of influenza (*Ferguson et al., 2003*; *Koelle et al., 2006*; *Bedford et al., 2012*). All three of these models, however, are subject to criticism. Influenza's epochal evolution model (*Koelle et al., 2006*) assumes that neutral or nearly neutral mutations accumulate at HA epitopes and that these changes enable a previously neutral mutation to exact a large antigenic effect and thereby to precipitate an antigenic cluster transition. Criticisms of this model are several. First, recent work indicates that amino acid changes that are responsible for cluster transitions have large antigenic effects in consensus sequences (*Koel et al., 2013*), such that genetic context is unlikely to be of utmost importance in determining the antigenic effect of mutations. Second, a molecular evolutionary analysis following the publication of the epochal evolution model has indicated that positive selection acts not only between antigenic clusters but also within antigenic clusters (*Suzuki, 2008*). In support of this finding, a more recent statistical analysis has shown that multiple antigenic variants co-circulate (*Strelkowa and Lässig, 2012*). Both of these empirical analyses are inconsistent with the assumptions of the epochal evolution model and indicate that the evolution of influenza is unlikely to be limited by the occurrence of antigenic mutations.

The two other existing models that can reproduce influenza's spindly HA phylogeny are the generalized cross-immunity model put forward by *Ferguson et al. (2003)* and the canalization model put forward by *Bedford et al. (2012)*. When simulated, both of these models yield antigenic variant co-circulation and are thus consistent with the analyses detailed above that have shown that influenza evolution is not antigenic mutation limited. In *Ferguson et al. (2003)*, generalized (strain-transcending) cross-immunity lasting on the order of 6 months was invoked to limit the genetic and antigenic diversity of influenza A/H3N2 in humans. The major criticism of this model is lack of empirical support for this duration and form of generalized cross-immunity: while there is evidence for long-lasting cross-immunity between more genetically distant strains (including heterologous influenza A subtypes), it appears to reduce pathology and possibly accelerate viral clearance rather than prevent infection (*Grebe et al., 2008*). A recent experimental study in ferrets indicates that generalized cross-immunity that prevents infection appears to exist, but that it lasts for less than a week (*Laurie et al., 2015*), which is insufficiently long to limit genetic and antigenic diversity in the model of Ferguson et al. The canalization model by Bedford et al. does not invoke generalized cross-immunity but instead assumes that antigenic mutations move viral strains in a two-dimensional (or higher) antigenic space. While these antigenic spaces or maps have been used to visualize the trajectories of flu's antigenic evolution (*Smith et al., 2004*; *de Jong et al., 2007*), models that start with the assumption of these maps considerably inflate the degree of competition between antigenic variants. This inflation of competition results from antigenic distances between daughter variants (variants that are produced from the same parent) necessarily being subadditive in this space. Inflated competition for susceptible hosts between antigenic variants is expected to lead to a more spindly phylogeny due to increased 'niche overlap' and therefore more frequent occurrences of between-strain competitive exclusion.

In light of our findings presented here and existing knowledge on the theory of asexual evolution, we can reflect on why the deleterious mutation and canalization models can reproduce limited genetic and antigenic diversity in simulations of influenza A/H3N2 in humans, whereas *Ferguson et al. (2003)* initially found that strain-specific immunity did not suffice in reproducing these patterns. From the population genetics literature on asexual populations in which interference effects are at play, it is well

known that an increase in the fitness variance of a population acts to slow adaptive evolution and make the characteristic size of adaptive mutations that fix larger. While here we have invoked deleterious mutations in the generation of fitness variation, beneficial mutations can similarly contribute to inherited variation in fitness (*Birky and Walsh, 1988*; *Barton, 1995*). Thus, in population genetic models, interference between beneficial mutations (or a combination of beneficial and deleterious mutations) similarly slows down the rate of adaptive evolution and increases the size of adaptive mutants that fix (*Birky and Walsh, 1988*; *Barton, 1995*; *Gerrish and Lenski, 1998*; *Rouzine et al., 2003*, *2008*; *Park et al., 2010*; *Sniegowski and Gerrish, 2010*; *Schiffels et al., 2011*; *Good et al., 2012*). However, all of these population genetic models assume a constant population size and therefore full resource competition between individuals. In the context of influenza, however, it is a change in antigenicity that is thought to provide the selective advantage. Because antigenic changes allow for escape from herd immunity, these changes by definition reduce competition for susceptible hosts (the virus resource). As such, antigenic mutants create a partially new niche and so do not necessarily lead to competitive exclusion. The establishment of antigenic mutants can therefore instead lead to long-term coexistence and, as a result of only partial cross-immunity, a larger infected population size. This is why explosive genetic and antigenic diversity is expected in the presence of only strain-specific immunity (*Ferguson et al., 2003*). In this context, generalized cross-immunity acts to considerably increase the competition between strains for susceptible hosts (as well as to reduce the overall infected population size). As such, the fitness variance generated by beneficial antigenic mutations in this model results in a decrease in the rate of antigenic evolution, an increase in the size of antigenic mutations that establish, and limited diversity in the long run, as in population genetic models that assume full competition between beneficial mutations by considering populations of constant size (*Good et al., 2012*). Similarly, the canalization model by Bedford et al. likely reproduces punctuated antigenic evolution and long-term limited genetic and antigenic diversity as a result of interference effects generated by inflated competition between antigenic mutations.

The generalized cross-immunity model, the canalization model, and the deleterious mutations model presented here therefore share fundamental similarities: they can reproduce the characteristic features of influenza evolution in humans by generating enough fitness variation among competing strains in the viral population. However, the first two of these models create fitness variation by beneficial antigenic mutations alone; because these mutations obtain their selective advantages by reducing competition for susceptible hosts, these models need other components (generalized cross-immunity or mutations that are necessarily subadditive in effect) to augment strain competition. In contrast, the deleterious mutation model we present here does not need to invoke processes to augment competition between antigenic strains because a large proportion of fitness variation in the virus population arises from differences in deleterious mutation loads (*Figure 7A*). Letting fitness variance be generated by deleterious mutations allows for limited diversity in the long run despite the co-circulation of antigenically very distinct variants that do not necessarily compete strongly for susceptible hosts. As such, our model can reproduce empirically observed high annual attack rates on the order of 10–15% (*Figure 10B* for $\lambda = 0.10$). Because deleterious mutations are known to circulate in the influenza A/H3N2 virus population (*Fitch et al., 1997*; *Pybus et al., 2007*; *Strelkowa and Lässig, 2012*), and in light of existing criticisms of the other two models, we therefore argue that the model presented here provides a more plausible mechanistic explanation for influenza's characteristic evolutionary features.

Our finding that specifically non-antigenic fitness variation is an important contributing driver in shaping the characteristic features of influenza's evolutionary dynamics in humans sheds light on other recent virological findings. One such finding is that cellular receptor binding avidity is an important phenotype that impacts viral fitness (*Hensley et al., 2009*). In a naive host, influenza viruses with low receptor binding avidities have a selective advantage, whereas, in a more immune host, influenza viruses with high receptor binding avidities have a selective advantage (*Hensley et al., 2009*). Which receptor binding avidity phenotype can be considered the 'deleterious' variant is therefore subject to which individual the virus finds itself in, as well as the overall degree of herd immunity in the host population (*Yuan and Koelle, 2013*). Because changes in receptor binding avidity frequently also alter antigenicity (*Hensley et al., 2009*), it is therefore also easily conceivable that certain of these changes increase virus fitness via two distinct mechanisms. As such, even though the genetic change is one and the same, the change in binding avidity that results can be thought of as increasing the background fitness of a virus strain, whereas the change in antigenicity that results can be considered as in this

paper. Successful cluster transitions via the 'jackpot' strategy, as depicted in *Figure 2C*, may therefore preferentially involve mutations that simultaneously affect binding avidity and antigenicity. Indeed, the majority of cluster-transition mutations that have been recently characterized fall near the receptor binding site of influenza's HA (*Koel et al., 2013*), suggesting that changes in the receptor binding avidity phenotype that occur with these antigenic mutations may improve virus fitness.

In addition to cellular receptor binding avidity, glycosylation of influenza's HA is known to impact virus fitness. Whether glycosylation sites accumulate over evolutionary time (as in the case of H3N2 in humans [*Blackburne et al., 2008*]) or do not (as in the case of H1N1 in humans [*Das et al., 2011*]) therefore will depend on how these sites impact the background fitness of virus strains. In the case of H1N1, for example, glycosylation has been shown to significantly reduce receptor binding avidity and therewith to lower overall virus fitness, despite the beneficial effect of glycosylation on escape from antibody-mediated neutralization (*Das et al., 2011*). Compensatory mutations are therefore needed to restore virus fitness following the addition of a glycosylation site (*Das et al., 2011*). Other phenotypes that impact virus fitness include those that influence protein stability (*Bloom and Glassman, 2009*) and those that confer resistance to antivirals (*Herlocher et al., 2004*). In the case of permissive mutations that influence protein stability (*Bloom et al., 2010*; *Gong et al., 2013*), these mutations may not directly influence virus fitness; their occurrence, however, may impact the viability of other mutations that have fitness consequences and thus may alter the size distribution of viable beneficial mutations, including those that allow for immune escape. Epistatic interactions such as these can therefore be accommodated within this general framework of fitness variation generated by phenotypes other than antigenicity in driving patterns of influenza's antigenic evolution. While we here simply model this fitness variation as arising from circulating deleterious mutations, any of these non-antigenic phenotypes can similarly contribute to this fitness variation. Indeed, deleterious mutations are necessarily deleterious as a consequence of some phenotype, whether it is susceptibility to antivirals, a suboptimal receptor binding avidity, a protein with reduced stability, or simply another phenotype that reduces viral replication. The complementary phenotypes to these can conversely be considered as beneficial mutations that contribute to fitness variation. As such, there is not always a clear source of fitness variation in the influenza virus population. What is critical, however, is that a significant proportion of the virus's fitness variation has to arise from non-antigenic phenotypes that do not reduce competition between virus strains, as antigenic mutants alone result in explosive genetic and antigenic diversity as a result of effective niche partitioning. Our choice of modeling simply deleterious mutation accumulation, instead of specific non-antigenic phenotypes, stems from the finding that virus populations, and more specifically influenza virus populations, carry substantial deleterious mutation loads (*Fitch et al., 1997*; *Pybus et al., 2007*).

Returning to our model, given that an antigenic mutation arises in a strain carrying some deleterious mutation load, one might expect the virus population's deleterious mutation load to increase in the long run, causing a long-term decline in influenza A/H3N2 fitness. Two processes exist, however, that can keep this long-term accumulation of deleterious mutations from occurring. First, as the virus population becomes less fit the proportion of non-antigenic mutations that are beneficial is likely to increase, such that the mutation-selection balance becomes an evolutionary attractor (*Goyal et al., 2012*). Second, within-subtype viral reassortment could occur sufficiently frequently to keep any long-term decline in viral fitness at bay by combining segments with low mutational load onto the same genetic background. The possibility that influenza A/H3N2 has been declining in fitness over time may, however, also be entertained: a recent virological analysis of human H3N2 viruses points towards a long-term decrease (since 1968) in the propensity of the virus to bind human sialic acid receptors (*Lin et al., 2012*). This decrease has been invoked to explain the reduction in this virus's disease impact over the last 10 years (*Lin et al., 2012*). Whether this finding can be interpreted as evidence for a 'weakening' of the virus over time is unclear, especially because 'weakening' by any epidemiological measure has not been empirically demonstrated. Clearly, more virological studies are needed to determine the fitness trajectory of influenza A/H3N2 in humans over the past decades.

While we focused on the role of deleterious mutations in shaping influenza A/H3N2's antigenic evolution, the presence of circulating deleterious mutations should also impact the adaptive evolutionary dynamics of other flu types/subtypes in humans. For example, influenza B is known to have a lower mutation rate than influenza A/H3N2 (*Nobusawa and Sato, 2006*; *Sanjuán et al., 2010*). Given that the deleterious mutation rate $\lambda$ is likely to decrease with the overall mutation rate, we would expect influenza B to carry a lower mean mutation load and further to have lower fitness

variance. As such, we would expect smaller antigenic mutants to be able to successfully establish (*Figure 1D*). Our model would therefore predict that influenza B evolves antigenically in a less punctuated manner than influenza A/H3N2—a pattern that has been recently documented (*Bedford et al., 2014*). All else equal, we would also expect influenza B to have a faster rate of antigenic evolution (*Figure 1E*). However, a lower mutation rate would also surely reduce the rate at which antigenic mutations occur. The slower rate of antigenic evolution observed for influenza B (*Bedford et al., 2014*) is therefore not inconsistent with our model. Relative to these two influenza viruses, human influenza A/H1N1 shows a similar pattern to H3N2 in terms of punctuated antigenic evolution (*Bedford et al., 2014*). Despite this similarity, H1N1's rate of antigenic evolution is considerably slower than H3N2's, although it is faster than influenza B's rate of antigenic evolution (*Bedford et al., 2014*, *2015*). The observed difference in rate of antigenic evolution between H3N2 and H1N1 is unlikely to reflect differences in these virus's mutation rates, since both are influenza A subtypes. Instead, these differences may stem from differences in their basic reproductive rates and, therefore, differences in selection pressures. This explanation is consistent with the finding that H1N1 appears to experience weaker antigenic selection than H3N2 in humans (*Bhatt et al., 2011*). Alternatively, the lower rate of antigenic evolution in H1N1 relative to H3N2 may be a consequence of differences in these viruses' global circulation patterns, which have only recently been described (*Bedford et al., 2015*).

Differences in the evolutionary dynamics of influenza viruses also exist across host species. For example, the same H3N2 virus that is circulating in humans emerged in pigs in the early 1970s. This swine influenza A/H3N2 evolves genetically at a rate that is similar to that of human influenza A/H3N2 (*de Jong et al., 2007*). However, its rate of antigenic evolution is approximately six times slower than the same virus's in humans (*de Jong et al., 2007*). This difference in the rate of antigenic evolution likely stems more from the stark ecological differences between the two hosts, rather than from differences in their deleterious mutation loads, with escape from humoral immunity being a less important evolutionary driver of HA in short-lived hosts than in humans.

Beyond the influenza viruses, circulation of deleterious mutations has been established in a wide range of RNA viruses (*Pybus et al., 2007*). The evolutionary dynamics of many of these viruses are characterized by spindly phylogenies and punctuated phenotypic changes. Whether deleterious mutations can account for these apparently similar patterns remains an open question. One especially intriguing case is that of norovirus, for which punctuated antigenic evolution has been documented (*Lindesmith et al., 2008*, *2011*) and for which deleterious mutations along with differential binding to histoblood group antigens may contribute to fitness variation (*Donaldson et al., 2008*; *Lindesmith et al., 2008*). Similar to the case of influenza, an interplay between antigenic and non-antigenic/deleterious mutations may alone be sufficient to explain this viral evolutionary pattern, obviating the need to invoke the mutation-limited process of epochal evolution (*Siebenga et al., 2007*; *Lindesmith et al., 2008*).

In addition to helping us understand patterns of viral adaptive evolution, acknowledging the 'rubbish around the ruby'—to paraphrase (*Peck, 1994*)—may also help us predict the course of adaptive evolution. Indeed, two recent publications have made great strides in predicting the genetic evolution of influenza A/H3N2 in humans over the short term (*Neher et al., 2014*; *Łuksza and Lässig, 2014*). Łuksza and Lässig's approach incorporated knowledge of antigenic and non-antigenic sites in developing a fitness model for this virus. With the assumption that mutations at non-antigenic sites were weakly deleterious, the authors were able to predict which influenza clades would grow and which ones would decline with an accuracy of 93% and 76%, respectively (*Koelle and Rasmussen, 2014*; *Łuksza and Lässig, 2014*). Instead of using knowledge specific to influenza A/H3N2, Neher et al.'s approach relied on branching patterns present in the HA phylogeny to predict strain evolution, with a success rate comparable to that of Łuksza and Lässig. The ability of both of these models to predict influenza evolution rests on the presence of substantial fitness variation in the influenza A/H3N2 virus population. The work here contributes to this understanding by reconciling the presence of fitness variation in the virus population with the observation of limited genetic and antigenic diversity of influenza A/H3N2's HA in the long run: cluster transitions will only succeed when large antigenic mutations find themselves in good genetic backgrounds, which must be largely determined by non-antigenic phenotypes. Successfully predicting cluster transitions will therefore require not only better characterizing the antigenic effects of mutations but also characterizing relative virus fitness, as mediated by differential deleterious mutation loads and contributions of non-antigenic phenotypes, in

influenza's HA and other gene segments. An integrative understanding of these non-antigenic components of viral fitness will require the continued work of virologists and modelers alike, and ideally their interaction, for predicting viral evolution.

## Materials and methods

### The deleterious mutation-selection balance in an epidemiological context

The deleterious mutation-selection balance was classically derived in the context of a discrete generation, constant-size Wright–Fisher population (*Haigh, 1978*). We here briefly re-derive the deleterious mutation-selection balance for a virus population in an explicit epidemiological context. To do so we consider a virus population subject exclusively to deleterious mutations. Because we are not considering antigenic variation at this point, the epidemiological dynamics are governed by a basic SIR model. In this model, the number of susceptible hosts increases only through births into the host population and decreases through background mortality and infection of susceptible hosts. The number of infected hosts increases through infection of susceptible hosts and decreases through recovery from infection and through natural mortality of infected hosts. The number of recovered hosts increases through recovery of infected individuals and decreases through natural mortality of recovered hosts. To extend this basic SIR model to allow for a virus population undergoing deleterious mutations, we classify infected individuals according to the number of deleterious mutations harbored by the virus they carry. This classification makes the implicit assumption that an infected host harbors a genetically homogeneous virus population; that is, in this case, that there is no variation in the number of deleterious mutations carried by distinct virions that make up the intrahost viral population. Although the assumption of a genetically homogeneous within-host virus population is clearly a simplifying assumption, it is an assumption that is commonly made in population-level models of influenza evolution (*Andreasen et al., 1997*; *Gog and Grenfell, 2002*; *Ferguson et al., 2003*; *Koelle et al., 2006*; *Bedford et al., 2012*). As in *Haigh (1978)*, we assume that mutations occur at birth, which, in the context of a virus population, are disease transmission events. We choose this approach to introduce new deleterious mutations over an approach that assumes that infected individuals can 'mutate' from being infected with a virus having a specified number of deleterious mutations to being infected with a virus having a higher number of deleterious mutations. This is because it is unlikely that a deleterious mutation that arises within a host could rapidly sweep to fixation in that host once the intrahost viral population is large. Introducing deleterious mutations at disease transmission events is therefore a more biologically reasonable assumption. Again as in *Haigh (1978)*, we further let the number of deleterious mutations that arise at 'birth' be Poisson distributed with mean $\lambda$, where $\lambda$ is the per-genome per-transmission deleterious mutation rate. With these assumptions, the rate of change in the number of infected individuals carrying a virus with $k$ deleterious mutations is given by:

$$\frac{dI_k}{dt} = \frac{S}{N} \sum_{j=0}^{k} \left( \beta_{k-j} e^{-\lambda} \frac{\lambda^j}{j!} I_{k-j} \right) - (\mu + \gamma) I_k,$$  (1)

where the first term in this equation captures the increase in the number of individuals infected with virus in mutation class $k$ arising from the transmission process and the second term captures the decrease in the number of infected individuals resulting from background mortality (at per capita rate $\mu$) and recovery from infection (at per capita rate $\gamma$). In the first term, $S$ is the number of susceptible hosts, $N$ is the (constant) host population size, and $\beta_i$ is the transmission rate of a virus carrying $i$ deleterious mutations. The Poisson term $e^{-\lambda} \frac{\lambda^j}{j!}$ provides the probability that $j$ deleterious mutations occur at transmission. As in *Haigh (1978)*, we assume multiplicative fitness effects of deleterious mutations, with each deleterious mutation exacting a fitness cost of size $s_d$. We can thus write $\beta_i$ in terms of the transmission rate of a virus in the highest fitness class (i.e., the lowest deleterious mutation class):

$$\beta_i = \beta_0 (1 - s_d)^i.$$  (2)

The dynamics of susceptible hosts are given by:

$$\frac{dS}{dt} = \mu N - \mu S - \frac{S}{N} \sum_{k=0}^{\infty} \beta_k I_k, \tag{3}$$

where the first term captures births into the host population (at a per capita rate $\mu$, equal to the background mortality rate), the second term captures background mortality of susceptible hosts, and the third term captures depletion of susceptible hosts through the transmission process. The dynamics of recovered individuals are simply given by $\frac{dR}{dt} = \gamma \sum_{k=0}^{\infty} I_k - \mu R$, where the first term captures the increase in the number of recovered individuals through recovery of infected individuals and the second term captures background mortality. For this set of equations, we can define the basic reproductive rate $R_{0,k}$ as the expected number of secondary infections (belonging to any mutation class) produced by a single infected individual carrying a virus with $k$ deleterious mutations in a completely susceptible population. This mutation class-specific basic reproductive rate is given by $R_{0,k} = \frac{\beta_k}{(\mu + \gamma)}$.

To solve for epidemiological equilibrium, we can now consider the dynamics of the first infected mutation class $k = 0$: $\frac{dI_0}{dt} = \beta_0 e^{-\lambda} \frac{S}{N} I_0 - (\mu + \gamma) I_0$. Setting this equation to zero, the equilibrium fraction of the population susceptible to infection can be solved for:

$$\frac{\widehat{S}}{N} = \frac{(\mu + \gamma)}{\beta_0 e^{-\lambda}}, \tag{4}$$

$\frac{\widehat{S}}{N}$ is greater than $\frac{1}{R_{0,k=0}}$ by a factor of $\frac{1}{e^{-\lambda}}$. That the fraction of susceptible hosts exceeds the inverse of the basic reproductive number of the highest-fitness virus class makes sense as the highest-fitness virus class occupies only a fraction of the total virus population.

The equilibrium number of infected individuals carrying virus with $k$ deleterious mutations, $\widehat{I}_k$, can now be solved. Substituting *Equations 2, 4* into *Equation 1* and simplifying yields:

$$\frac{dI_k}{dt} = (\mu + \gamma) I_{tot} \sum_{j=0}^{k} \left( (1 - s_d)^{k-j} \frac{\lambda^j}{j!} p_{k-j} \right) - (\mu + \gamma) I_k,$$

where the total number of infected individuals is given by $I_{tot} = \sum_{i=0}^{\infty} I_i$ and $p_i$ is the proportion of infected individuals carrying virus with $i$ deleterious mutations, $p_i = \frac{I_i}{I_{tot}}$. Setting this equation to 0 and solving for $p_k$ yields:

$$p_k = \sum_{j=0}^{k} \left( (1 - s_d)^{k-j} \frac{\lambda^j}{j!} p_{k-j} \right).$$

This equation mirrors *Equation 4* in reference (*Haigh, 1978*), with solution:

$$p_k = e^{-\theta} \frac{\theta^k}{k!}, \tag{5}$$

where $\theta = \lambda/s_d$. The total number of infected individuals at equilibrium, $\widehat{I}_{tot}$, can now be solved by substituting *Equations 2, 4* into *Equation 3*, and replacing $I_k$ with $p_k I_{tot}$. Setting equal to 0 and solving for $I_{tot}$ yields:

$$\widehat{I}_{tot} = \frac{e^{-\lambda} \mu N \left( 1 - \frac{\mu + \gamma}{\beta_0 e^{-\lambda}} \right)}{(\mu + \gamma) \sum_{k=0}^{\infty} (1 - s_d)^k p_k}.$$

Substituting *Equation 5* into the above expression and simplifying yields:

$$\widehat{I}_{tot} = \frac{\mu N}{(\mu + \gamma)} \left( 1 - \frac{\mu + \gamma}{\beta_0 e^{-\lambda}} \right). \tag{6}$$

The equilibrium number of mutation class-specific infected individuals can then be calculated using $\widehat{I}_k = p_k \widehat{I}_{tot}$ for any deleterious mutation class $k$.

Absolute fitness in epidemiological models is provided by the net reproductive rate $R$. With a population experiencing deleterious mutations, each mutation class $k$ will have its own net reproductive rate $R_k$ at equilibrium. Given the above definition of the basic reproductive rate for viruses carrying $k$ deleterious mutations, the fitness cost associated with deleterious mutations (*Equation 2*), and the equilibrium fraction of susceptible hosts (*Equation 4*), the mutation class $k$ net reproductive rate is given by:

$$R_k = R_{0,k}\left(\frac{\widehat{S}}{N}\right) = e^{\lambda}(1-s_d)^k.$$ (7)

As expected, the mean net reproductive rate of the virus population ($\sum_{k=0}^{\infty}(R_k p_k)$) is 1 when the population is at epidemiological equilibrium, with some viruses having a net reproductive rate above 1 and other viruses having a net reproductive rate below 1. The variance in the net reproductive rate is given by $\sum_{k=0}^{\infty}(R_k^2 p_k) - 1$, which increases with increases in $\lambda$ and increases with decreases in $s_d$.

## Initial and final reproductive rates of an antigenic mutant evolving de novo from a resident viral population

To compute the antigenic mutant's initial and final net reproductive rates, we first use an epidemiological model to mathematically describe the interaction between the resident antigenic strain and the antigenic mutant. We denote the resident strain with super- and subscripts $r$ and the antigenic mutant with super- and subscripts $m$, and use a history-based model (*Andreasen et al., 1997*) to specify the immunological interaction between these two strains. Note here that we again make the implicit assumption that the intrahost viral population is genetically homogeneous with respect to both antigenicity and the number of deleterious mutations carried. With deleterious mutations accumulating at transmission, we have:

$$\frac{dS_0}{dt} = \mu N - \mu S_0 - \frac{S_0}{N}\sum_{k=0}^{\infty}\left(\beta_k\left(I_{0,k}^r + I_{B,k}^r + I_{0,k}^m + I_{A,k}^m\right)\right),$$

$$\frac{dI_{0,k}^r}{dt} = \frac{S_0}{N}\sum_{j=0}^{k}\left(\beta_{k-j}e^{-\lambda\frac{\lambda^j}{j!}}\left(I_{0,k-j}^r + I_{m,k-j}^r\right)\right) - (\mu+\gamma)I_{0,k}^r,$$

$$\frac{dI_{0,k}^m}{dt} = \frac{S_0}{N}\sum_{j=0}^{k}\left(\beta_{k-j}e^{-\lambda\frac{\lambda^j}{j!}}\left(I_{0,k-j}^m + I_{r,k-j}^m\right)\right) - (\mu+\gamma)I_{0,k}^m,$$

$$\frac{dS_r}{dt} = \gamma\sum_{k=0}^{\infty}I_{0,k}^r - \mu S_r - \sigma\frac{S_r}{N}\sum_{k=0}^{\infty}\left(\beta_k\left(I_{0,k}^m + I_{r,k}^m\right)\right),$$ (8)

$$\frac{dS_m}{dt} = \gamma\sum_{k=0}^{\infty}I_{0,k}^m - \mu S_m - \sigma\frac{S_m}{N}\sum_{k=0}^{\infty}\left(\beta_k\left(I_{0,k}^r + I_{m,k}^r\right)\right),$$

$$\frac{dI_{m,k}^r}{dt} = \sigma\frac{S_m}{N}\sum_{j=0}^{k}\left(\beta_{k-j}e^{-\lambda\frac{\lambda^j}{j!}}\left(I_{0,k-j}^r + I_{m,k-j}^r\right)\right) - (\mu+\gamma)I_{m,k}^r,$$

$$\frac{dI_{r,k}^m}{dt} = \sigma\frac{S_r}{N}\sum_{j=0}^{k}\left(\beta_{k-j}e^{-\lambda\frac{\lambda^j}{j!}}\left(I_{0,k-j}^m + I_{r,k-j}^m\right)\right) - (\mu+\gamma)I_{r,k}^m,$$

$$\frac{dS_{\{r,m\}}}{dt} = \gamma\sum_{k=0}^{\infty}\left(I_{r,k}^m + I_{m,k}^r\right) - \mu S_{r,m},$$

where $S_x$ is the number of uninfected individuals who have previously been infected with strain(s) $x$, and $I_{x,k}^y$ is the number of individuals previously infected with strain $x$ who are currently infected with a virus of strain $y$ carrying $k$ deleterious mutations. The parameter $\sigma$ quantifies the extent to which susceptibility to infection with one strain is affected if the individual has previously been infected with the other strain. With $\sigma = 1$ a previous infection does not reduce susceptibility to infection with the second strain, whereas with $\sigma = 0$ a previous infection results in complete protection from reinfection with a second infection. A higher value of $\sigma$ therefore corresponds to a greater degree of immune

escape (i.e., a larger antigenic change). As expected, at the time immediately prior to the antigenic mutant's emergence, *Equation 8* reduce to *Equations 1, 3*.

Given *Equation 8*, an antigenic mutant that arises in a background with $i$ deleterious mutations initially has a net reproductive rate of:

$$R_m(t=0) = \left(\frac{\beta_i}{\mu+\gamma}\right)\left(\frac{S_0}{N} + \sigma\frac{S_r}{N}\right),$$ (9)

where we have defined time $t$ in terms of the time since the antigenic mutant's emergence.

With $\frac{\widehat{S_0}}{N} = \frac{(\mu+\gamma)}{\beta_0 e^{-\lambda}}$ (*Equation 4*) and with $\frac{\widehat{S_r}}{N} \approx 1 - \frac{\widehat{S_0}}{N}$, the mutant's initial net reproductive rate is given by:

$$R_m(t=0) = e^\lambda(1-s_d)^i\left(1 + \sigma\left(\frac{\beta_0 e^{-\lambda}}{\mu+\gamma} - 1\right)\right).$$ (10)

This expression would be exact if we allowed for coinfection, treating individuals currently infected with the resident strain similarly to individuals previously infected with the resident strain. *Equation 10* consists of three components: (i) $e^\lambda$ is the net reproductive rate of resident strain viruses that are in mutational class $k = 0$; (ii) $(1 - s_d)^i$ quantifies the extent to which the antigenic mutant's initial reproductive rate is reduced as a result of the deleterious mutations it carries; and (iii) the term $\left(1 + \sigma\left(\frac{\beta_0 e^{-\lambda}}{\mu+\gamma} - 1\right)\right)$ quantifies the extent to which the antigenic mutant's initial reproductive rate is increased as a result of immune escape. To make the link stronger between this model and traditional population genetic models, we can define the selective advantage of an antigenic mutant at the time of its emergence as $s_b = \sigma\left(\frac{\beta_0 e^{-\lambda}}{\mu+\gamma} - 1\right)$.

The reproductive rate of a strain carrying an antigenic mutation necessarily decreases as it establishes through its own accumulation of deleterious mutations. Neglecting any changes in the host immune landscape, the final mean reproductive rate of the antigenic mutant lineage is:

$$R_m(t=\infty) = e^\lambda\left(\sum_{j=i}^\infty p_{j-i}(1-s_d)^j\right)\left(1 + \sigma\left(\frac{\beta_0 e^{-\lambda}}{\mu+\gamma} - 1\right)\right),$$ (11)

where the second term of the product quantifies the extent to which the antigenic mutant lineage's reproductive rate is reduced as a result of the deleterious mutations it carries once it has reached its own deleterious mutation-selection balance. Because this term ignores the possibility of stochastic loss of mutation class $i$ during the strain's establishment, *Equation 11* is an upper estimate for $R_m(t = \infty)$. Changes in the host immune landscape would change the third term of the product.

Under this simple model, successful establishment can only occur when $R_m(t = \infty) > 1$. From *Equation 11*, it is therefore clear that the probability of an antigenic mutant's establishment is higher the lower its original mutation load $i$. This is because, in the absence of any compensatory or back-mutations, $i$ provides the lower limit to the deleterious mutation load carried by the new antigenic strain. It is also clear from this equation that the probability of an antigenic mutant's establishment is higher the greater the ability of the mutant strain to reinfect previously infected individuals, reflected in a higher $\sigma$.

## Model parameters

Two evolutionary parameters determine influenza's deleterious mutation load: the per-genome per-transmission deleterious mutation rate $\lambda$ and the transmission fitness cost of a deleterious mutation $s_d$. Neither of these parameters have been estimated specifically for influenza. We therefore do our best with estimating them using existing data from other viruses or via rough estimates that incorporate existing knowledge about influenza. The parameter $\lambda$ was roughly computed by first multiplying the genome size of influenza's major coding regions (12,741 nucleotides [*Holmes et al., 2005*]) with the empirical estimate of $2.3 \times 10^{-5}$ for the number of substitutions that occur per nucleotide per cell infection in influenza A viruses (*Sanjuán et al., 2010*). This yielded a per-genome, per-transmission substitution rate of 0.293. With roughly 2/3 of mutations being non-synonymous and roughly 50% of non-lethal, non-synonymous mutations being deleterious (*Sanjuán et al., 2004*), we calculated the per-genome, per-transmission deleterious mutation rate $\lambda$ as the product: $0.293 \times \frac{2}{3} \times 0.5 \approx 0.10$. Given the uncertainty in $\lambda$, we look across a range of $\lambda$ values from $\lambda = 0$ to $\lambda = 0.20$ in *Figure 1D,E*.

To get a rough estimate of the transmission fitness cost of deleterious mutations for influenza, we start with empirical estimates of mean fitness cost values for the five viruses studied in *Sanjuán (2010)*: 0.103, 0.107, 0.112, 0.126, and 0.132. These fitness costs are in vitro fitness costs associated with viral growth in cells. These costs therefore differ from transmission fitness costs at the population level, which is how $s_d$ is defined in the models we present. To obtain a rough estimate for transmission fitness cost from these within-host viral growth fitness costs, we used a published within-host model of influenza dynamics (*Baccam et al., 2006*) to simulate viral load dynamics under two scenarios: a 'wild-type' parameterization and a 'deleterious mutant' parameterization, where the mutant carries a specified fitness cost. The 'wild-type' scenario used parameter values that were estimated in *Baccam et al. (2006)* using viral load data from six individuals who were experimentally infected with influenza. To parameterize the 'deleterious mutant' scenario, we assumed that the in vitro fitness cost manifested itself through a reduction in the within-host viral production rate. We therefore set all of the within-host parameters of the 'deleterious mutant' scenario to be equal to the ones used in the 'wild-type' scenario, with the exception of the viral production rate. This parameter we set to the product of the 'wild-type' scenario's viral production rate and the quantity 1 minus the in vitro fitness cost. For the six patients in *Baccam et al. (2006)*, we simulated the 'wild-type' and the 'deleterious mutant' scenarios for each of the five in vitro fitness costs listed above. We then mapped these simulated viral load dynamics to between-host transmission rates by assuming that viral transmission rates are proportional to the log of viral load: $\beta(\tau) \propto log_{10}(V(\tau))$. This assumption is commonly used (see *Handel et al., 2013*) and has some empirical support, particularly from studies of HIV (*Quinn et al., 2000*; *Fraser et al., 2007*). The population-level transmission fitness costs $s_d$ for the five empirical within-host fitness cost estimates from *Sanjuán (2010)* could thus be inferred for each of the patients studied in *Baccam et al. (2006)*. Our simulations indicate, first and foremost, that relatively large fitness costs within a host translate into much smaller fitness effects at the population level. This is in part unsurprising given that we assume that transmission rates scale with the log of the viral load. We further found that in vitro fitness costs in the last three patients studied in *Baccam et al. (2006)* would translate into fitness benefits in terms of transmission. Rather than trusting this to be the case biologically, this paradoxical result is likely to be a result of the rough mapping that we (and others) use for translating between viral load and transmissibility. We therefore restricted ourselves to the first three patients studied in *Baccam et al. (2006)*. For them, we found that within-host fitness costs of 0.103–0.132 yielded transmission fitness costs between 0.002 and 0.018, with a mean transmission fitness cost $s_d$ of 0.008, which we use in *Figure 1A–C*. Given the uncertainties present in our estimate of $s_d$, we instead show three different values for $s_d$ in *Figure 1D,E*. These indicate that the qualitative effects of circulating deleterious mutations (larger average antigenic sizes of mutants that establish and slower rates of antigenic evolution) are robust to specific values for $s_d$.

In addition to these two evolutionary parameters, the models we present contain three epidemiological parameters: the birth/death rate $\mu$, the recovery rate $\gamma$, and the basic reproductive rate of a virus in the $k = 0$ mutation class ($R_{0,k=0}$). We use a birth/death rate of $\mu = 1/30$ years$^{-1}$, based on crude birth rate estimates of approximately 33 per 1000 individuals per year (*United Nations, 2011*). This value for $\mu$ is also consistent with a recent flu modeling study (*Bedford et al., 2012*). The recovery rate $\gamma$ was chosen based on an in-depth meta-analysis of 56 human challenge studies (*Carrat and Flahault, 2007*). In this study, the authors found an average duration of viral shedding of 4.8 days for influenza viruses (regardless of subtype). They also calculated an average generation time of 3.1 days for influenza A/H3N2, where generation time is defined as the time between an individual becoming infected and transmitting the virus. Because, in an epidemiological model with a constant transmission rate and a constant recovery rate, the duration of infectiousness (taken to be the duration of viral shedding) and the generation time take the same value when a virus is endemically circulating, we chose an intermediate value between these estimates, letting the recovery rate $\gamma = 1/4$ days$^{-1}$. We chose an $R_{0,k=0}$ of 2.25. This value falls within the range of recent $R_0$ estimates for influenza A/H3N2 (range = 1.21–3.58 for influenza A/H3N2's second pandemic wave [*Jackson et al., 2010*]).

*Figure 1D,E* require specification of a distribution for the degree of immune escape $\sigma$. We assume that $\sigma$ is gamma distributed with mean of 0.012 and a shape parameter of 2. The choice of a gamma distribution is based on a virological study showing that the distribution of beneficial mutation effects appear to be gamma distributed (*Sanjuán et al., 2004*). Our choice of mean $\sigma$ value is not based on an independent empirical estimate as relevant data to parameterize this value to our knowledge do not exist. Assuming an exponential distribution with a mean of 0.012 for $\sigma$ instead of a gamma distribution

did not change the qualitative findings that circulating deleterious mutations act to slow antigenic evolution and make it more punctuated in nature.

*Figures 2, 3* in the main text show stochastic simulations of the epidemiological model given by *Equation 8*. To simulate the model stochastically, we used Gillespie's $\tau$-leap method with a time step of 1 hr. These stochastic simulations required further specification of a host population size $N$. We used a population size of $N$ = 4 billion hosts, corresponding to the human population size around 1980, which can also be roughly considered today's tropical population size.

## Description and implementation of the phylodynamic model

To explore the full evolutionary dynamics of our model with non-antigenic and antigenic mutations, we implemented an individual-based model that allowed for an arbitrary number of new strains to enter the population and co-circulate. Individuals in this model were categorized as either currently infected or currently uninfected. Using the same notation as in history-based multi-strain models (*Andreasen et al., 1997*), each currently uninfected individual carries a list of antigenic types with which he has previously been infected. Each infected individual similarly carries this list of previously experienced antigenic types. In addition to this list, each infected individual carries a current viral infection that is characterized by the infecting virus's deleterious mutation load and its antigenic type. Again, we assume for the sake of model simplicity that the intrahost viral population is genetically homogeneous such that a single deleterious mutation load and a single antigenic type suffice in characterizing a viral infection. Upon recovery, infected individuals add to their strain history list the antigenic type of the viral infection from which they are recovering.

Viral antigenic types are defined by their relationship to one another in terms of their pairwise cross-immunity values $\sigma$. The minimum antigenic distance between the challenging strain and the host's repertoire of previously encountered strains (as provided by the host's antigenic type list) determines the probability of a currently uninfected host becoming infected by the challenging strain. Specifically, the probability of infection given contact with a host infected with strain $i$ was given by $p_{ij} = \min(1.0, \sigma_{ij})$, where strain $j$ is the strain in the host's repertoire that is antigenically closest to strain $i$. This formulation leads to complete immunity to reinfection with antigenic types that a host has previously experienced and complete susceptibility to infection for naive hosts. To compute any $\sigma_{ij}$ value, the model uses tracked parent–offspring relationships of distinct antigenic variants, with mother–daughter variants differing by one antigenic mutation having a degree of cross-immunity that is drawn from the specified antigenic size distribution (see below). The degree of cross-immunity $\sigma_{ij}$ between two strains $i$ and $j$ is assumed to be additive; for example, if strain $i$ gives rise to strain $l$ and strain $l$ gives rise to strain $j$, then $\sigma_{ij} = \sigma_{il} + \sigma_{lj}$.

When they occurred, both antigenic and non-antigenic mutations occurred during transmission events. A virus in an infected host therefore never changed antigenically or changed in mutation load during the course of infection. At a transmission event, the number of non-antigenic mutations was drawn from a Poisson distribution with mean $\lambda$. Each of the non-antigenic mutations was characterized as beneficial (with probability $\varepsilon_c$) or deleterious (with probability $1 - \varepsilon_c$), where $\varepsilon_c \ll 1$. The net change in the number of deleterious mutations was then calculated and the virus's mutation load was appropriately increased or decreased. Our phylodynamic simulations required that a proportion of non-antigenic mutations were beneficial rather than deleterious because of the finite number of hosts we were computationally able to simulate. Specifically, in finite populations it is well known that the lowest-load mutation classes will at some point be stochastically lost, initiating Muller's ratchet (*Muller, 1964*). Recent work has indicated, however, that if a fraction of mutations are beneficial rather than deleterious, the average mutation load in a population can be maintained indefinitely over time (*Goyal et al., 2012*). In our simulations, we set $\varepsilon_c$ to a value of 0.16, which resulted in a stable deleterious mutation load over time.

The number of antigenic mutations occurring at a transmission event was drawn from a Poisson distribution with mean $\lambda_{antigenic}$, independently of the number of non-antigenic mutations. We used a $\lambda_{antigenic}$ of 0.00075. The size of each of these antigenic mutations was drawn from a gamma distribution with a mean of 0.012 and a shape parameter of 2. The antigenic difference $\sigma_{ij}$ between the virus $i$ in the infecting host and the virus $j$ in the host becoming infected was calculated as the sum of the sizes of the antigenic mutations occurring at transmission.

The simulations were run using an individual-based stochastic simulation algorithm based on Gillespie's tau-leap algorithm, using a time step $\tau$ of 1 day. Experimentation with smaller time steps yielded similar results, such that we used a $\tau$ of 1 day for computational efficiency. Over each time

interval $\tau$, the number of events occurring during that interval was drawn from a Poisson distribution. The modeled events were births, deaths, recoveries, and infectious contacts. Individuals born into the population were considered naive to infection (such that their strain history lists were empty). Deaths occurred from both uninfected and currently infected hosts, indiscriminately. For each infected host death, an infected individual was randomly chosen to be discarded from the population, independent of the virus currently infecting the individual. Similarly, for each recovery, an infected individual was chosen at random to recover. For each infectious contact from individuals infected with a class-$k$ virus, an infected individual was chosen at random from the current class-$k$ infected population and a currently uninfected individual was also chosen at random. The uninfected host was then infected with probability $p_{ij}$, where, as described above, $p_{ij}$ is given by $\min(1.0, \sigma_{ij})$, where $\sigma_{ij}$ is the antigenic distance between infecting strain $i$ and the strain $j$ in the host's repertoire that is antigenically closest to strain $i$. We did not allow for coinfection, so our phylodynamic simulations did not include the possibility of viral reassortment leading to antigenic cluster transitions. For these individual-based simulations we used a population size of $N = 40$ million individuals, 1/100th of the human population size in 1980. Our choice of this limited population size was due to the large amount of memory required to keep track of the immune histories of each individual host. The relationships of who-infected-whom and at what time were tracked such that the true phylogenetic tree could be reconstructed and compared against influenza A/H3N2's HA phylogeny reconstructed from empirical sequences.

Preliminary simulations revealed highly volatile population dynamics, with large peaks in prevalence followed by extinction or long periods of low prevalence. To stabilize the dynamics we allowed a small number of infectious contacts to occur from outside of the focal population by having uninfected individuals experience an additional force of infection from an external pool of infected individuals ($M = 200$ for all simulations). So as not to alter the evolutionary dynamics, this external pool of infected individuals was assumed to have the same mutational load distribution and antigenic type frequencies as the focal population.

All remaining evolutionary and epidemiological parameters used in these individual-based simulations were the same as the ones listed in the model parameterization section above. The simulations were implemented in Java using a modified version of the program Antigen (http://bedford.io/projects/antigen/). The original code was modified to track the mutation load and the antigenic phenotypes of the viral population, which in our case requires us to track the pairwise distances between all antigenic types instead of viral locations on a two-dimensional antigenic surface. Source code for our full phylodynamic model is available on the GitHub repository at https://github.com/davidrasm/MutAntiGen.git.

In our simulations, we tracked the fitness of individual viruses in terms of their net reproductive rate $R$. For a particular virus $i$ carrying $k$ deleterious mutations,

$$R = \frac{\beta_k}{(\mu + \nu)} \frac{S_{eff}}{N},$$

where $S_{eff}$ is the effective number of susceptible hosts available to a particular virus. Because the susceptibility of a host to a particular virus depends on the host's detailed immune history, to compute $S_{eff}$ we first compute the probability $\rho_{ij}$ of strain $i$ infecting each host in the population upon contact, and then sum $\rho_{ij}$ over all uninfected hosts in the population to arrive at $S_{eff}$. In practice, for computationally tractability, we only sum over a large number ($n = 10,000$) of randomly sampled hosts to approximate $S_{eff}$. Given the distribution of $R$ in the viral population, we can then compute the fraction of viral fitness variation attributable to deleterious mutations (i.e., variation in $\beta_k$) vs antigenic differences among strains (i.e., variation in $S_{eff}$). To decompose the total variance in fitness into these components, we first log-transform $R$, so that

$$\ln(R) = \ln(\beta_k) + \ln\left(\frac{S_{eff}}{N}\right) + \ln\left(\frac{1}{(\mu + \nu)}\right).$$

Dropping the constant $\frac{1}{(\mu + \nu)}$ term, we can then decompose the total variance in log $R$ into its component parts using the well-known relation that the variance of the sum of two random variables is equal to the sum of their variances plus their covariance,

$$var(R) = var(\beta_k) + var\left(\frac{S_{eff}}{N}\right) + 2cov\left(\beta_k, \frac{S_{eff}}{N}\right).$$

Note that the covariance term cannot be ignored because $\beta_k$ and $\frac{S_{eff}}{N}$ are not independent random variables and can be correlated due to the shared common ancestry of viral strains (i.e., phylogenetic correlations). To compute the fraction of total viral fitness variation attributable to $\beta_k$ and thus deleterious mutations, we subtract the variance attributable to the covariance:

$$\text{var explained by } \beta_k = \frac{var(\beta_k)}{var(R) - 2cov\left(\beta_k, \frac{S_{eff}}{N}\right)}.$$

We also conducted additional simulations that did not include any antigenic evolution but did include deleterious mutations to determine if the spindly phylogeny shown in *Figure 6* arose simply due to the presence of purifying selection rather than being driven by antigenic change. Simulations without antigenic evolution were conducted with the same evolutionary and epidemiological parameters; average viral $R_0$ was as in the simulations with antigenic evolution except $\lambda_{\text{antigenic}}$ was set to zero. However, because there was no antigenic evolution, the average number of infected humans over time and the annual attack rates were significantly lower than in the simulations with antigenic evolution. To compensate for this, in *Figure 9*, we allowed immunity to wane over time, as in standard SIRS epidemiological models. The rate of immune waning was set such that immunity from a previous infection lasted 12 years on average before the host became completely susceptible again. This rate of immune waning was chosen to produce an average annual attack rate of ~5–10%, consistent with what was observed in the simulations with antigenic and non-antigenic mutations.

## Acknowledgements

We thank Chris Illingworth, Alexei Drummond, and three anonymous reviewers for insightful feedback on this work.

## Additional information

### Funding

| Funder | Grant reference | Author |
| --- | --- | --- |
| James S. McDonnell Foundation (JSMF) | | Katia Koelle, David A Rasmussen |
| National Institute of General Medical Sciences (NIGMS) | U54-GM111274 | Katia Koelle |
| U.S. Department of Homeland Security | RAPIDD program of the Science and Technology Directorate | Katia Koelle |

The funders had no role in study design, data collection and interpretation, or the decision to submit the work for publication.

### Author contributions

KK, Conception and design, Acquisition of data, Analysis and interpretation of data, Drafting or revising the article; DAR, Acquisition of data, Analysis and interpretation of data, Drafting or revising the article

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
