## [Decision Letter]

Thank you for sending your work entitled “The effects of a deleterious mutation load on patterns of influenza A/H3N2's antigenic evolution in humans” for consideration at *eLife*. Your article has been favorably evaluated by Ian Baldwin (Senior Editor) and three reviewers, one of whom served as Guest Reviewing Editor.

The paper by Koelle and Rasmussen treats the effects of deleterious mutations in the evolution of the influenza virus. This is an important aspect of influenza evolution, because traditional theory and data analysis of influenza evolution has focused on antigenic effects alone and neglected other evolutionary forces such as the genetic load caused by deleterious mutations. The dynamics analysed in this paper are complex, because they are shaped by the interplay of beneficial antigenic mutations, deleterious background mutations, and by large fluctuations in population size and absolute fitness caused by the epidemiology of the virus. The main findings are that deleterious mutations (a) change the speed and character of antigenic evolution, and (b) affect the epidemiology of influenza; in particular, they can explain the linear shape of the influenza tree.

Explaining the elegant ladder-like (‘spindly’) phylogenetic pattern of H3N2 evolution in humans has been a prize aim of infectious disease modelers for at least a decade. The authors present a thought-provoking model that emphasizes the importance of considering key antigenic substitutions within the context of a high background deleterious mutation load. The need for a very fit mutation (that causes a large antigenic change) to overcome a deleterious background also explains the punctuated nature of antigenic changes.

Some parts of the article are quite difficult to follow, and it is not clear if it is possible to reproduce the results without further explanation. Please provide a more thorough description of the model.

Essential revisions:

1) Connection to the theory of asexual evolution:

The results in the first part of the paper should be put in context of the general theory of asexual evolution. In light of this theory, the result that deleterious mutations reduce the probability of fixation for beneficial mutations is well known. This effect is contained in the theory of background selection (e.g. in the work of B. Charlesworth), as well as in travelling wave models: deleterious mutations generate fitness variance, which makes the fixation probability of a beneficial mutation strongly background-dependent (Good et al., Distribution of fixed beneficial mutations and the rate of adaptation in asexual populations, PNAS 2012). Beneficial mutations with effect size below a certain threshold fix with a reduced rate close to neutral mutations, which results in an increased average effect of fixed mutations (Good et al., PNAS 2012; Schiffels et al., Emergent neutrality in adaptive asexual evolution, Genetics 2011); specifically for influenza, the reduction of the adaptive rate due to linked deleterious mutations in non-adaptive sequence has also been observed in [72].

It would also be good to discuss the possible consequences of epistasis in this context (if applicable).

2) Stressing the interplay between evolution and epidemiology:

In view of 1), it would be good to focus the paper more on the interplay between the evolutionary and the epidemiological dynamics, which is the novel and most interesting part of the paper. In particular, the authors find that the reduced rate and increased average effect of antigenic mutations may contribute to the linear shape of the influenza tree and to increased attack rates. However, it should become clearer that a strong pruning of antigenic mutations by the joint dynamics with deleterious mutations requires that the latter contribute a substantial fraction of the average fitness variance in the population. This may be the case in model simulations (and the results on tree shape should be compared with the relative contributions of mutation classes to fitness variance). But it is not clear for the actual system. Furthermore, we suggest the authors juxtapose their finding of the influence of deleterious mutations on tree shape with previous explanations of this characteristic.

3) The effect of deleterious mutations:

In the Discussion, the authors argue that the presence of deleterious mutations reduce clonal competition compared to populations without mutational load. We think this argument is incorrect. According to all models cited above, deleterious mutations should add to the fitness variance in the population and, hence, increase the amount of interference selection. This does not contradict the suggested pruning of antigenic mutations, because in the joint model, antigenic competition is no longer equivalent to fitness competition. The antigenic variance in the population may decrease with increasing rate or effect of deleterious mutations, even if the total fitness variance increases.

4) Connection to empirical observations and previous models:

One concern is that the model is not particularly data-driven. Although empirical trees are presented in Figure 4 to illustrate 3 pathways to antigenic change that are consistent with the model, these trees are more useful as illustrative of concepts than as good evidence for the model. The authors should stress the illustrative character of that figure.

As for (virtually) all theoretical models, it is possible that the patterns in the trees could alternatively be explained by other mechanisms. For example, in the BE92 to WU95 cluster transition, the 145N to 145K substitutions that did not take off globally (that are dead-ends) could have occurred in locations that are thought to not be global source regions (i.e. could the failure of those viruses to persist globally despite beneficial mutations relate to their emergence in less inter-connected non-source regions (i.e., other than SE Asia))? A more thorough discussion of the limits of the model would be in place. For example, in the WU95-SY97 transition, would a scenario not predicted by the model be a longer branch length? How long?

It's useful to explore how robust a flu model is to other subtypes and hosts, and the authors make an admirable attempt to consider their model in the context of influenza B and swine. However, the most natural comparison would be with seasonal H1N1, and its omission is striking here. The authors also mistakenly assume that H3N2 has similar mutation rates in different hosts, and as a result miss the opportunity to examine how well their model works in a system such as swine (a particularly good example because versions of the human H3N2 virus also circulates in pigs) which have evolutionary rates that are higher than in humans for non-surface proteins (Worobey paper) and should carry a higher mutational load than humans (Discussion, third paragraph). The swine comparison raises ecological considerations that may also relate to human demography.

In the Discussion, it would be important to describe in more detail if the competing models have fundamental issues in capturing the current knowledge or if they would be valid for other parameter choices. It seems odd to write “2%” (Discussion, first paragraph) for a model that is not described in detail and that has presumably free parameters.

5) Technical issues:

In general, it is difficult to follow the mathematical description of the model. The description is not particularly complete. For example, in [Disp-formula equ3]: Why does the general background death rate lead to a positive term for the change in the number of susceptible hosts? Or is it only the background death rate of infected individuals? Why does recovery not lead to a positive term here? Another example is the definition of *R*_*0,k*_
*=0*, which differs from the inverse of [Disp-formula equ5] only by a factor in the exponential function, or [Disp-formula equ13], which is not a mere modification of [Disp-formula equ6], but an extension.

The statement in the last paragraph of the subsection “Model parameters” is problematic. [Disp-formula equ14] is a set of (deterministic) ODEs and thus does not give a stochastic model. It is well known that many stochastic models can be the basis of a single ODE model, so there is no unique model that the authors simulate. Are the rates in [Disp-formula equ14] directly used, i.e. for a Gillespie algorithm? Or do they reflect net rates that describe the net effect of several rate processes?

[Editors' note: further revisions were requested prior to acceptance, as described below.]

Thank you for resubmitting your work entitled “The effects of a deleterious mutation load on patterns of influenza A/H3N2's antigenic evolution in humans” for further consideration at *eLife*. Your revised article has been favorably evaluated by Ian Baldwin (Senior Editor), a Reviewing editor, and two reviewers. The manuscript has been improved but there are some remaining issues that need to be addressed before acceptance, as outlined below:

In the revised version, the authors have successfully addressed most of the comments. In particular, they have substantially improved the population-genetic side of their findings, which was one of the main concerns of the reviewers.

However, we ask the authors to be more explicit about some of the assumptions in their model. The success or failure of a beneficial antigenic mutation is a complex process, and while the authors do not need to incorporate all aspects into their model, they should be explicit about what is not being factored in:

1) The authors refer to selection as occurring on a ‘strain’, whereas in reality mutations are occurring within a swarm (quasi species) of intrahost diversity. The frequency of mutation as well as back-mutation within this swarm means that beneficial and deleterious mutations are not only arising but also being removed. The impact of a mutation on viral fitness is therefore also a product of the frequency of that mutation within the swarm. The potential bottleneck at transmission also means that not all variants, particularly low-frequency variants, will be transmitted between hosts.

2) We believe the authors are also assuming in this model a consistent immune landscape, for simplicity, which does not reflect variation in immune responses and prior exposures among hosts that affect the fitness of antigenic mutations.

Host variation can have a profound effect on the fitness of a particular mutation (as evidenced nicely by the antigenic mutations in receptor binding sites, which have very different fitnesses in hosts with different immune profiles based on whether antigenic escape or binding avidity is more important.

3) Successful H3N2 antigenic variants most frequently are produced in Southeast Asia before spreading globally. The spatial ecology of the virus has pronounced effects on which mutations succeed on a global scale and which are not sustained. It would have been simple to examine the proportion of early 145K mutations that arose in SE Asia on the phylogeny (suggested in the original reviewer comments), but short of that the importance of spatial ecology in the global success of a particular variant should be discussed (no additional analysis is required).

4) The authors should clarify that/why beneficial non-antigenic mutations were not included in the model.

5) There is still a lack of clarity and the role of competition between antigenic variants in the model. We find it hard to agree with the argument that 'rather, we propose that many of these circulating antigenic variants ultimately decline from the accumulation of deleterious mutations in the context of an only slowly changing herd immunity landscape”.

---

## [Author Response]

*Essential revisions*:

*1) Connection to the theory of asexual evolution*:

*The results in the first part of the paper should be put in context of the general theory of asexual evolution. In light of this theory, the result that deleterious mutations reduce the probability of fixation for beneficial mutations is well known. This effect is contained in the theory of background selection (e.g. in the work of B. Charlesworth), as well as in travelling wave models: deleterious mutations generate fitness variance, which makes the fixation probability of a beneficial mutation strongly background-dependent (Good et al., Distribution of fixed beneficial mutations and the rate of adaptation in asexual populations, PNAS 2012). Beneficial mutations with effect size below a certain threshold fix with a reduced rate close to neutral mutations, which results in an increased average effect of fixed mutations (Good et al., PNAS 2012; Schiffels et al., Emergent neutrality in adaptive asexual evolution, Genetics 2011); specifically for influenza, the reduction of the adaptive rate due to linked deleterious mutations in non-adaptive sequence has also been observed in*
[72].

*It would also be good to discuss the possible consequences of epistasis in this context (if applicable)*.

We agree that situating the results of the first part of the paper in the context of the theory of asexual evolution would greatly improve the paper. To this end, we have first added the following text to the Introduction: “We start by extending classic population genetic models into an epidemiological context; as expected from previous analyses of these types of models (5; 11; 23; 55; 57), we find that circulating sublethal deleterious mutations in influenza A/H3N2’s viral population reduce the rate of adaptive evolution and increase the average size of the beneficial mutants that fix.”

Under the Results section, we have now fully situated the results shown in Figure 1 in the context of the theory of asexual evolution, and also cited [72]: “Given a specified size distribution for antigenic mutations […] This leads to a reduced tempo of antigenic change, consistent with a reduction in the rate of adaptation that is known from the population genetics literature (5; 57).”

Finally, in the Discussion section, we now very briefly mention the consistency of epistasis with the model we present.

*2) Stressing the interplay between evolution and epidemiology*:

*In view of 1), it would be good to focus the paper more on the interplay between the evolutionary and the epidemiological dynamics, which is the novel and most interesting part of the paper. In particular, the authors find that the reduced rate and increased average effect of antigenic mutations may contribute to the linear shape of the influenza tree and to increased attack rates. However, it should become clearer that a strong pruning of antigenic mutations by the joint dynamics with deleterious mutations requires that the latter contribute a substantial fraction of the average fitness variance in the population. This may be the case in model simulations (and the results on tree shape should be compared with the relative contributions of mutation classes to fitness variance). But it is not clear for the actual system. Furthermore, we suggest the authors juxtapose their finding of the influence of deleterious mutations on tree shape with previous explanations of this characteristic*.

This is again an excellent comment. We now first highlight the importance of epidemiological dynamics and the competition-reducing nature of antigenic change in the Introduction: “Gaining intuition from these simple models […] a pattern that is inconsistent with the long-term evolutionary dynamics of influenza A/H3N2 in humans.”

In Figure 2, which is the first model we present that is explicitly a multi-strain epidemiological model, we further emphasize the importance of considering the interplay between evolutionary and epidemiological dynamics, and also added the following excerpt to the Results: “The above model contains a number of simplifying assumptions […] population genetic models that assume a constant population size may not be appropriate under certain epidemiological conditions.”

In Figure 6, we have also added several subplots that show more clearly that in our model deleterious mutations contribute a substantial fraction of fitness variance in the viral population. In particular, Figure 6 shows that between 20-80% of the fitness variance is explained by deleterious mutations. How this proportion is calculated is detailed in added text under Materials and methods. Figure 6 (which shows the deleterious mutation load distributed on viral lineages present in the phylogeny) further illustrates that deleterious mutations contribute substantially to fitness variance. This figure can be compared to the added Figure 6, which shows the distribution of the net reproductive rate (i.e., absolute fitness) on the same viral lineages.

In the Discussion, we now also in detail review the three published epidemiological models that can reproduce the spindly HA tree ([40], [7], and [22]), describing in detail their shortcomings and how their findings can be interpreted in the context of the findings presented here.

For the first model (40), we have the following text: “Influenza’s epochal evolution model (40) assumes that neutral or nearly neutral mutations accumulate at HA epitopes […] and indicate that the evolution of influenza is unlikely to be limited by the occurrence of antigenic mutations.”

For the latter two models, which both reproduce antigenic co-circulation, we have added the following passage: “The generalized cross-immunity model, the canalization model, and the deleterious mutations model presented here therefore share fundamental similarities […] and in light of existing criticisms of the other two models, we therefore argue that the model presented here provides a more plausible mechanistic explanation for influenza’s characteristic evolutionary features.”

*3) The effect of deleterious mutations*:

*In the Discussion, the authors argue that the presence of deleterious mutations reduce clonal competition compared to populations without mutational load. We think this argument is incorrect. According to all models cited above, deleterious mutations should add to the fitness variance in the population and, hence, increase the amount of interference selection. This does not contradict the suggested pruning of antigenic mutations, because in the joint model, antigenic competition is no longer equivalent to fitness competition. The antigenic variance in the population may decrease with increasing rate or effect of deleterious mutations, even if the total fitness variance increases*.

This comment reflects different interpretations of the term ‘clonal interference’ in the literature. If ‘clonal interference’ is used to capture all interference effects (both positive and negative), we completely agree with the reviewers. In the original submission, we were using the term ‘clonal interference’ to refer exclusively to interference caused by co-circulating beneficial mutants (as in the work by Gerrish and Lenski). We have clarified this throughout the text, using the term ‘interference effects’ over ‘clonal interference’. We also now emphasize in the Discussion that what appears to be needed to constrain antigenic diversification is that a substantial fraction of fitness variance is generated by deleterious mutations or phenotypes *unrelated to antigenicity*.

*4) Connection to empirical observations and previous models*:

*One concern is that the model is not particularly data-driven. Although empirical trees are presented in*
Figure 4
*to illustrate 3 pathways to antigenic change that are consistent with the model, these trees are more useful as illustrative of concepts than as good evidence for the model. The authors should stress the illustrative character of that figure*.

We have stressed the illustrative character of Figure 4 even more than before, indicating that the observed phylogenetic patterns shown in Figure 4
*are consistent with* the proposed molecular pathways of antigenic cluster transitions.

*As for (virtually) all theoretical models, it is possible that the patterns in the trees could alternatively be explained by other mechanisms. For example, in the BE92 to WU95 cluster transition, the 145N to 145K substitutions that did not take off globally (that are dead-ends) could have occurred in locations that are thought to not be global source regions (i.e. could the failure of those viruses to persist globally despite beneficial mutations relate to their emergence in less inter-connected non-source regions (i.e., other than SE Asia))? A more thorough discussion of the limits of the model would be in place. For example, in the WU95-SY97 transition, would a scenario not predicted by the model be a longer branch length? How long*?

We have added text mentioning that the observed patterns in Figure 4’s trees could also be explained by other mechanisms. For example, for Figure 4, we now have this paragraph: “Our models above indicate that the failure of these early 145K […] would be necessary to confirm our genetic background hypothesis.”

*It's useful to explore how robust a flu model is to other subtypes and hosts, and the authors make an admirable attempt to consider their model in the context of influenza B and swine. However, the most natural comparison would be with seasonal H1N1, and its omission is striking here. The authors also mistakenly assume that H3N2 has similar mutation rates in different hosts, and as a result miss the opportunity to examine how well their model works in a system such as swine (a particularly good example because versions of the human H3N2 virus also circulates in pigs) which have evolutionary rates that are higher than in humans for non-surface proteins (Worobey paper) and should carry a higher mutational load than humans (Discussion, third paragraph). The swine comparison raises ecological considerations that may also relate to human demography*.

We appreciate that the referees are interested in how deleterious mutations would further impact the evolutionary dynamics of H1N1 in humans and H3N2 in pigs. We kept this section brief in our first submission, but now extend it further. For seasonal H1N1 in humans, we have added the following paragraph: “Relative to these two influenza viruses, human influenza A/H1N1 shows a similar pattern to H3N2 […] a consequence of differences in these viruses’ global circulation patterns, which have only recently been described (9).”

We have also added text on swine influenza H3N2, referencing de Jong et al.’s finding that the rate of genetic evolution of this virus’s HA is similar to that of human influenza A/H3N2, while its rate of antigenic evolution is 6x slower. Worobey et al.’s 2014 Nature paper infers a significantly higher substitution rate for swine than for humans. Neither gets at differences in *l* well, though, because both quantify substitution rates, not mutation rates. (The difference between these rates is described nicely in Belshaw, Sanjuan, & Pybus (2011) Current Opinion in Virology.) We now also mention the importance of ecological factors (host lifespan) in generating different selective pressures on the HA in humans vs. pigs (please see the following passage: “Differences in the evolutionary dynamics of influenza viruses also exist across host species. […] being a less important evolutionary driver of HA in short-lived hosts than in humans.”

*In the Discussion, it would be important to describe in more detail if the competing models have fundamental issues in capturing the current knowledge or if they would be valid for other parameter choices. It seems odd to write “2%” (Discussion, first paragraph) for a model that is not described in detail and that has presumably free parameters*.

We agree with this comment and have altered the text appropriately. We now discuss both the Bedford paper and the Ferguson paper in the context of these models creating fitness variance solely by the co-circulation of beneficial antigenic mutants. Generalized cross-immunity (in the Ferguson paper) and subadditive antigenic effects (in the Bedford paper) both augment competition between circulating beneficial antigenic mutants, and thus result in the generation of spindly HA phylogenies. We removed mention of any specific attack rates.

*5) Technical issues*:

*In general, it is difficult to follow the mathematical description of the model. The description is not particularly complete. For example, in*
[Disp-formula equ3]*: Why does the general background death rate lead to a positive term for the change in the number of susceptible hosts? Or is it only the background death rate of infected individuals? Why does recovery not lead to a positive term here? Another example is the definition of* R_0,k_ =0*, which differs from the inverse of*
[Disp-formula equ5]
*only by a factor in the exponential function, or*
[Disp-formula equ13]*, which is not a mere modification of*
[Disp-formula equ6]*, but an extension*.

We have reworked the entire Materials and methods section entitled ‘The deleterious mutation-selection balance in an epidemiological context’ to address this concern. This section now details the specific terms in the differential equations, and reorganizes the analytical calculations such that they are more straightforward to follow.

To address the specific questions here: The positive term *mN* in the equation for the rate of change of susceptible hosts reflects births into the host population (the birth rate is equal to the death rate, and both are quantified by *m*, such that the host population remains constant in size. Recovery from infection does not lead to a positive term in the equation for the rate of change of susceptible hosts because individuals recover into the recovered (and immune) class. This first model is without antigenic evolution, so is given by an SIR (not SIRS) model. The definition of *R*_*0,k*_*=0* is now more straightforward because we now ignore lethal deleterious mutations (such that rho_L is 0).

*The statement in the last paragraph of the subsection “Model parameters” is problematic.*
[Disp-formula equ14]
*is a set of (deterministic) ODEs and thus does not give a stochastic model. It is well known that many stochastic models can be the basis of a single ODE model, so there is no unique model that the authors simulate. Are the rates in*
[Disp-formula equ14]
*directly used, i.e. for a Gillespie algorithm? Or do they reflect net rates that describe the net effect of several rate processes*?

We simulated the model stochastically using the Gillespie’s *t*-leap algorithm, and have added text to that effect.

[Editors' note: further revisions were requested prior to acceptance, as described below.]

*In the revised version, the authors have successfully addressed most of the comments. In particular, they have substantially improved the population-genetic side of their findings, which was one of the main concerns of the reviewers*.

We again thank the referees for suggesting that we place our findings in the context of population genetic theory on asexual evolution. We believe that this has considerably improved our manuscript.

*However, we ask the authors to be more explicit about some of the assumptions in their model. The success or failure of a beneficial antigenic mutation is a complex process, and while the authors do not need to incorporate all aspects into their model, they should be explicit about what is not being factored in*:

*1) The authors refer to selection as occurring on a ‘strain’, whereas in reality mutations are occurring within a swarm (quasi species) of intrahost diversity. The frequency of mutation as well as back-mutation within this swarm means that beneficial and deleterious mutations are not only arising but also being removed. The impact of a mutation on viral fitness is therefore also a product of the frequency of that mutation within the swarm. The potential bottleneck at transmission also means that not all variants, particularly low-frequency variants, will be transmitted between hosts*.

We fully agree with this comment. In recent years, there have been an increasing number of studies that examine influenza’s intrahost diversity and attempt to quantify the size of influenza’s transmission bottleneck (Varble et al. (2014) Cell Host Microbe, Murcia et al. (2010) J. Virology, Murcia et al. (2012) PLoS Pathogens, Hoelzer et al. (2010) J. Virology, Illingowrth et al. (2014) PLoS Computational Biology, Ghedin et al. (2011) J. Infectious Diseases, Hughes et al. (2012) PLoS Pathogens, among others). These include both experimental and observational studies, and consider influenza infections of both human and non-human animal hosts. While these studies are exceptionally valuable, no mathematical models that focus on understanding influenza evolution at the population-level over the long-term have to date integrated this extent of genetic complexity into their model simulations. Instead, the most well known, as well as the most recent, multi-strain models of influenza at the population level have all – without exception that we know of – made the model-simplifying assumption that hosts, when infected, are infected with a single genotype of the virus (e.g. [27] PNAS, [1] J. Math Bio., [22] Nature, [40] Science, [7] BMC Biology and papers based on this model, most recently [9] Nature). We similarly make this modeling assumption. Although not ‘true’, this assumption greatly simplifies the structure of these types of dynamical systems models. Whether explicitly including intrahost diversity in any of these population-level models would make any difference in terms of predicted dynamics should be theoretically considered at some point in a separate piece of work. The degree of realism that is incorporated into epidemiological models is always to some extent a subjective decision, although it should be based in large part on the question being addressed in the work.

In our manuscript, the fundamental question we address is the role that circulating deleterious mutations play in shaping influenza’s antigenic evolution. What is critical in terms of the model is therefore not how faithfully we model the exact manner in which deleterious mutations arise or are removed from the influenza viral population (this will depend on the size of the transmission bottleneck, the degree of intrahost diversity, and the relationship between within- host and population-level fitness). Instead, *what is critical is that the formulation of our model allows for the transient circulation of deleterious mutations in influenza’s viral population*. This transient circulation of deleterious mutations has been empirically established through phylogenetic analyses of influenza virus sequence data ([24] PNAS, [59] Molecular Biology and Evolution). While a highly complex model that incorporates intrahost genetic diversity and transmission bottlenecks of a specified size may more faithfully capture the mechanistic origins of deleterious mutations, this model’s complexity would very likely limit our ability to identify the specific roles that circulating deleterious mutations play in shaping patterns of influenza’s antigenic evolution. Furthermore, there are still too many unknowns related to intrahost influenza diversity and the size of the transmission bottleneck to even accurately include these complexities in model simulations.

While our model does to some extent simplify the manner in which deleterious mutations mechanistically arise and are removed from the influenza viral population, we make the most biologically reasonable assumptions possible given this simplification. Specifically, we assume that deleterious mutations arise at the time of transmission and that they are removed from the population with recovery events of infected individuals. We make the assumption that deleterious mutations arise at the time of transmission to be consistent with population genetic models which assume that deleterious mutations arise at birth. Presumably, deleterious mutations do arise during the period of infection. Indeed, all of the multi-strain influenza models referenced above assume that individuals ‘mutate’ from being infected with one genotype/antigenic phenotype of the virus to being infected with a different genotype/antigenic phenotype of the virus (e.g., I_1_ → I_2_). While this formulation may make sense for antigenic mutants, which are likely to have a selective advantage within infected hosts, this formulation is not reasonable for deleterious mutations. This is because if transmission-reducing deleterious mutations emerged de novo within hosts during their infectious period, it is unlikely that they would experience a strong intrahost selective advantage. As such, they would not rise to appreciable frequencies within hosts. Simulating the origination of deleterious mutations as coming from infected individuals ‘mutating’ from being infected with one viral strain carrying *k* deleterious mutations to being infected with a lower-fitness strain (in terms of transmission) carrying *k+1* or more deleterious mutations is therefore not a great modeling choice. We therefore have deleterious mutations be introduced at the time of transmission (‘birth’). For consistency, we similarly model the introduction of antigenic mutations at transmission in the phylodynamic model. As the referees have noted, the assumption of a single genotype within an infected host is not biologically accurate (‘true’). However, *short of modeling intrahost genetic diversity and transmission bottlenecks explicitly, we feel that the formulations of our models adopt the most parsimonious assumptions possible in terms of when mutations accumulate*.

Although we strongly feel that intrahost genetic diversity or transmission bottlenecks do not need to be explicitly modeled in our submitted work, we now explicitly state our assumption of a genetically homogeneous intrahost viral population (i.e., we explicitly state that we do not incorporate intrahost influenza genetic diversity in our models). We have made this clarification under the Materials and methods section entitled ‘The deleterious mutation-selection balance in an epidemiological context’ (“To extend this basic SIR model to allow for a virus population undergoing deleterious mutations […] is the per-genome per-transmission deleterious mutation rate”).

Under the Materials and methods section entitled ‘Initial and final reproductive rates of an antigenic mutant evolving de novo from a resident viral population’, we have also added: “We denote the resident strain with super- and subscripts *r* and the antigenic mutant with super- and subscripts *m*, and use a history-based model (1) to specify the immunological interaction between these two strains. Note here that we again make the implicit assumption that the intrahost viral population is genetically homogeneous with respect to both antigenicity and the number of deleterious mutations carried.”

Finally, under the Materials and methods section entitled ‘Description and implementation of the phylodynamic model’, we have added the sentence: “Again, we assume for the sake of model simplicity that the intrahost viral population is genetically homogeneous such that a single deleterious mutation load and a single antigenic type suffice in characterizing a viral infection.”

We have further re-read the entirety of the manuscript to look for instances in which we explicitly refer to selection occurring on a ‘strain’, but have not been able to identify any such instances. We do refer to competitive exclusion of strains, as well as between-strain cross- immunity. This is terminology that is commonly used in epidemiological models of influenza; removing this terminology would significantly reduce the clarity of the presented work.

*2) We believe the authors are also assuming in this model a consistent immune landscape, for simplicity, which does not reflect variation in immune responses and prior exposures among hosts that affect the fitness of antigenic mutations*.

*Host variation can have a profound effect on the fitness of a particular mutation (as evidenced nicely by the antigenic mutations in receptor binding sites, which have very different fitnesses in hosts with different immune profiles based on whether antigenic escape or binding avidity is more important*.

We are unsure of what the reviewers exactly mean by ‘a consistent immune landscape’. In the phylodynamic model (which we assume is the specific model being referenced in this comment), individuals can and do differ in their infection histories. We model these infection histories in the same way as standard, frequently-used history-based epidemiological models infection histories (see Andreasen et al. (1997*)* J. Math Bio.). One individual in the population at time *t* may have been infected with antigenic strain types 1, 2, and 3, for example (each strain also carrying a certain number of deleterious mutations). Another individual at time *t* may have been infected with only antigenic strain type 2. A third individual at time *t* may not have been previously infected with any influenza strain (i.e. is considered completely naive). When challenged with a strain through contact with an infected individual, individuals will become infected with a probability that depends on their infection history, as detailed under the ‘Description and implementation of the phylodynamic model*’* section of the manuscript. Specifically, the following (already existing) text describes this process: “The model tracked the complete immune history of each host by recording the antigenic type of all previous infections […] The degree of cross-immunity *s*ij between two strains *i* and *j* is assumed to be additive, e.g., if strain *i* gives rise to strain *l* and strain *l* gives rise to strain *j*, then *s*ij = *s*il + *s*_lj_.”

We do not include individual variation in host immune responses beyond variability in susceptibility due to infection history. *Given contact with an infected individual, previous infection history alone determines susceptibility to infection in our model. The number of deleterious mutations harbored by the virus an individual is infected with alone affects the infectivity of the infected individual* (i.e., the number of productive contacts made per unit time that may lead to infection). Both of these model specifications are already detailed under the Materials and methods section ‘Description and implementation of the phylodynamic model*’*. Neither the infection history nor the number of deleterious mutations harbored by the virus an individual is carrying affects the infected individual’s recovery rate. These are all modeling choices we made in an effort to adopt biologically reasonable assumptions. The modeling choice of previous infection history affecting a host’s susceptibility to infection upon challenge with a new strain is furthermore based on existing epidemiological influenza models present in the literature ([1] J. Math Bio., [22] Nature, [40] Science, [7] BMC Biology) which make use of this same assumption. Because our submitted work specifically considers the impact of deleterious mutations in shaping patterns of influenza’s antigenic evolution, we do not explicitly incorporate receptor binding avidity. Our first revision, however, does go into considerably detail on how this phenotype can be considered in the context of the model we present. Specifically, in the Discussion section, we have the following text: “Our finding that specifically non-antigenic fitness variation is an important contributing driver in shaping the characteristic features of influenza’s evolutionary dynamics in humans sheds light on other recent virological findings […] suggesting that changes in the receptor binding avidity phenotype that occur with these antigenic mutations may improve virus fitness.”

For increased emphasis, we have revised the Materials and methods section ‘Description and implementation of the phylodynamic model*’* to ensure that the readers know that host differ in their immune histories in this full model (please see the following passage: “To explore the full evolutionary dynamics of our model with non-antigenic and antigenic mutations, we implemented an individual-based model […] infected individuals add to their strain history list the antigenic type of the viral infection from which they are recovering.”

*3) Successful H3N2 antigenic variants most frequently are produced in Southeast Asia before spreading globally. The spatial ecology of the virus has pronounced effects on which mutations succeed on a global scale and which are not sustained. It would have been simple to examine the proportion of early 145K mutations that arose in SE Asia on the phylogeny (suggested in the original reviewer comments), but short of that the importance of spatial ecology in the global success of a particular variant should be discussed (no additional analysis is required)*.

First, we would like to apologize to the referees for not addressing this point to the extent that was desired. We are of course familiar with the work that demonstrates that the spatial ecology of the virus has pronounced effects on which mutations succeed on a global scale and the importance of E/SE Asia (and now also India) on the sourcing of antigenic variants ([64] Science, [45] PLoS Pathogens, [9] Nature, among several others). The comments that we were provided in the first round of reviews on this point stated:

*Connection to empirical observations and previous models*:

*One concern is that the model is not particularly data-driven. Although empirical trees are presented in*
Figure 4
*to illustrate 3 pathways to antigenic change that are consistent with the model, these trees are more useful as illustrative of concepts than as good evidence for the model. The authors should stress the illustrative character of that figure*.

*As for (virtually) all theoretical models, it is possible that the patterns in the trees could alternatively be explained by other mechanisms. For example, in the BE92 to WU95 cluster transition, the 145N to 145K substitutions that did not take off globally (that are dead-ends) could have occurred in locations that are thought to not be global source regions (i.e. could the failure of those viruses to persist globally despite beneficial mutations relate to their emergence in less inter-connected non-source regions (i.e., other than SE Asia))? A more through discussion of the limts of the model would be in place. For example, in the WU95-SY97 transition, would a scenario not predicted by the model be a longer branch length? How long*?

We addressed this point in our first revision by stressing the illustrative character of Figure 4 and adding text that mentions that the observed patterns in Figure 4’s trees could also be explained by other mechanisms. With specific reference to the BE92-to-WU95 cluster transition, we added the following text: “Our models above indicate that the failure of these early 145K […] would be necessary to confirm our genetic background hypothesis.”

Given the reviewers’ comments on our original submission (reproduced above), we did not recognize that they were requesting that we examine the proportion of early 145K mutations that arose in SE Asia on the phylogeny. We have now performed a straightforward additional analysis to examine the geographic locations of the early HA sequences that carried the 145K mutation to determine whether or to what extent spatial ecology may have instead played in the failure of these early 145K lineages.

Of the 395 full-length HA nucleotide sequences in NCBI’s Influenza Virus Resource Database that were isolated between 1993 and 1997 (the date range used for Figure 4, which examined the BE92-to-WU95cluster transition), 60% were from the US. Only 15% were from Asia (China, Hong Kong, Malaysia, Japan, Singapore). Another 15% were from Europe. The remaining 10% were largely from Oceania. Given such incredibly low sampling from Asian countries, it becomes somewhat meaningless to attempt to do any phylogeographic analyses on the early WU95-like 145K clades that did not succeed. That said, we nevertheless looked at the three largest unsuccessful WU95-like 145K clades present in Figure 4. The largest of these three clades had 10 sequences. 7 of the sequences were from the US (ranging in geographic extent from Louisiana to Massachusetts, with no state represented more than once). 1 sequence came from Canada. Two sequences came from Europe (Spain and England). Although none of these sequences were from Asia, this does not mean that this clade did not circulate in Asia. (Remember, only 15% of the available sequences came from Asia, so Asia was heavily undersampled.) What this analysis does show is that this clade was geographically widespread, with circulation in both the US and Europe.

The second largest clade had 4 sequences, with Germany, W. Virginia, Finland, and Singapore as the geographic locations of these 4 sequences. The third largest clade had 3 sequences, with Nebraska, Alaska, and Kwangju (South Korea) as the geographic locations of these 3 sequences. Both of these clades were therefore also geographically widespread.

Given these findings, we have added text to the main manuscript in the Results section that alludes to Figure 4.

*4) The authors should clarify that/why beneficial non-antigenic mutations were not included in the model*.

In the simpler models (Figures 1, 2, 3 and 5), we did not include beneficial non-antigenic mutations. This is because the primary question our manuscript seeks to address is how deleterious mutations can shape patterns of influenza’s antigenic evolution. By including deleterious non- antigenic mutations, we here show that they can greatly impact influenza’s antigenic dynamics. Including beneficial non-antigenic mutations would not alter this result. This is because, similarly to deleterious mutations, beneficial mutations increase fitness variance. Increasing fitness variance in the virus population will make the fate of antigenic mutants even more context-dependent. Please see the following passage in the Discussion: “From the population genetics literature on asexual populations in which interference effects are at play […] The establishment of antigenic mutants can therefore instead lead to long-term coexistence and, as a result of only partial cross-immunity, a larger infected population size.”

Later in the Discussion, we also have written: “While we here simply model this fitness variation as arising from circulating deleterious mutations, any of these non-antigenic phenotypes can similarly contribute to this fitness variation […] influenza virus populations carry substantial deleterious mutation loads (24; 59).”

We believe these paragraphs make clear that including beneficial non-antigenic mutations would not alter our qualitative results, and also clarifies why we focus on deleterious mutations in this work.

Note also that in the more complex phylodynamic model (Figures 6, 7, 8, 9 and 10) we did include beneficial non-antigenic mutations, to avoid Muller’s ratchet. We introduced these mutations through a parameter *e*, which specified the proportion of non-antigenic mutations that were beneficial rather than deleterious. We added this complexity because it is well known that in a finite population, the most fit genotype can be (repeatedly) lost through a process called Muller’s ratchet, thereby initiating a long-term decrease in viral fitness. We have rewritten the Materials and methods section accordingly (“At a transmission event, the number of non-antigenic mutations was drawn from a Poisson distribution with mean *l* […] which resulted in a stable deleterious mutation load over time.”

5) There is still a lack of clarity and the role of competition between antigenic variants in the model. We find it hard to agree with the argument that “rather, we propose that many of these circulating antigenic variants ultimately decline from the accumulation of deleterious mutations in the context of an only slowly changing herd immunity landscape”.

In response to this concern, we have examined in greater detail the reasons for loss of the antigenically distinct clades shown in (new) Figure 6. All of these clades (unsurprisingly) have reproductive values that transition from having a net reproductive rate above 1 to having a net reproductive rate below 1 (new Figure 6), such that they are excluded. For these clades, we examined their loss of fitness as brought about by deleterious mutation accumulation (new Figure 6), and we further examined their loss of fitness as brought about by a decline in their number of susceptible hosts (new Figure 6). Some of these clades did not see a precipitous decline in their number of susceptible hosts, such that their decline in fitness was almost entirely attributed to deleterious mutation accumulation (e.g., the large clade originating in year 20 and going extinct in year 28). For other clades, however, the picture was less clear, and both a decline in the number of susceptible hosts and an increase in the deleterious mutation load were evident. We rephrased the text accordingly (please see the passage: “In our model, these antigenic variants […] suffice in generating a sufficiently high-fitness viral lineage that will ensure its long-term evolutionary success”).

Thus, our inclusion of the new Figure 6, alongside Figure 6, makes it more evident that antigenic evolution and purifying selection act in concert with one another in shaping influenza’s phylodynamics.